# A protective role for *Drosophila* Filamin in nephrocytes via Yorkie mediated hypertrophy

Sybille Koehler[1,2] ⬤, Tobias B Huber[2], Barry Denholm[1] ⬤

Podocytes are specialized epithelial cells of the kidney glomerulus and are an essential part of the filtration barrier. Because of their position, they are exposed to constant biomechanical forces such as shear stress and hydrostatic pressure. These forces increase during disease, resulting in podocyte injury. It is likely podocytes have adaptative responses to help buffer against deleterious mechanical force and thus reduce injury. However, these responses remain largely unknown. Here, using the *Drosophila* model, we show the mechanosensor Cheerio (dFilamin) provides a key protective role in nephrocytes. We found expression of an activated mechanosensitive variant of Cheerio rescued filtration function and induced compensatory and hypertrophic growth in nephrocytes depleted of the nephrocyte diaphragm proteins Sns or Duf. Delineating the protective pathway downstream of Cheerio we found repression of the Hippo pathway induces nephrocyte hypertrophy, whereas Hippo activation reversed the Cheerio-mediated hypertrophy. Furthermore, we find Yorkie was activated upon expression of active Cheerio. Taken together, our data suggest that Cheerio acts via the Hippo pathway to induce hypertrophic growth, as a protective response in abnormal nephrocytes.

## Introduction

Podocytes are highly specialized epithelial cells in the kidney glomerulus, which together with fenestrated endothelial cells and the glomerular basement membrane form the three-layered filtration barrier. The unique morphology of podocytes is exemplified by the formation of primary and secondary foot processes. The adjacent foot processes of neighbouring podocytes interdigitate completely enwrapping the glomerular capillaries. This specialized arrangement facilitates the formation of a unique cell–cell contact, the slit diaphragm (SD) (Arakawa, 1970; Pavenstädt et al, 2003). Interestingly, the glomerular filtrate has the highest extravascular flow rate in the human body and depends on the hydrostatic

pressure difference across the filtration barrier (Kriz & Lemley, 2017). Podocyte morphology and attachment to the glomerular basement membrane is crucial to counteract these forces of flow and pressure.

Because of their position and function in the glomerular filter, podocytes face continual and fluctuating biomechanical forces of different types such as tensile forces induced by hydrostatic pressure in the capillaries and fluid shear stress in the filtration slit as well as in the Bowman's space (Endlich et al, 2017). These biomechanical forces increase during diseases such as hypertension and diabetes, inflicting podocyte injury that results in changes to their morphology, detachment from the basement membrane, and ultimately to their loss as they are shed into the primary urine. As post-mitotic cells, podocytes cannot be replenished, which leaves the capillaries blank and result in proteinuria, loss of protein from blood into urine. To avoid this, it is conceivable that podocytes possess mechanisms to monitor and adapt to fluctuations in biomechanical forces, which are up-regulated upon podocyte injury. Indeed, they contain a contractile actin-based cytoskeleton similar to that in smooth muscle cells suggesting their morphology is "plastic" which might be an important component of the adaptive mechanisms required to withstand mechanical forces (Faul et al, 2007). This actin-based cytoskeleton plays a crucial role in the development and maintenance of podocyte morphology.

In general, the process of transducing biomechanical force into biochemical signal is called mechanotransduction (Endlich et al, 2017). Mechanosensors—as they detect biomechanical force—are essential for this process. Several classes of mechanosensors such as ion channels, proteins associated with the actin-cytoskeleton, and proteins of the extracellular matrix, are already known (Endlich et al, 2017). In fact, a few proteins were already described to have a mechanosensor function in podocytes. Among them are the ion channels TRPC6 and P2X4 (Anderson et al, 2013; Forst et al, 2016) as well as the actin-associated proteins Cofilin, Paxillin, and Filamin (Koukouritaki et al, 1999; Endlich et al, 2001; Ashworth et al, 2010; Garg et al, 2010; Greiten et al, 2021; Okabe et al, 2021). However, detailed investigations into the functional roles of the mechanosensors in podocytes are scarce. In addition, the question whether

---

[1]Biomedical Sciences, University of Edinburgh, Edinburgh, UK   [2]III Department of Medicine, University Medical Center Hamburg-Eppendorf, Hamburg, Germany

Correspondence: Sy.koehler@uke.de
Sybille Koehler's present address is Campus Forschung N27, UKE Hamburg, Hamburg, Germany.

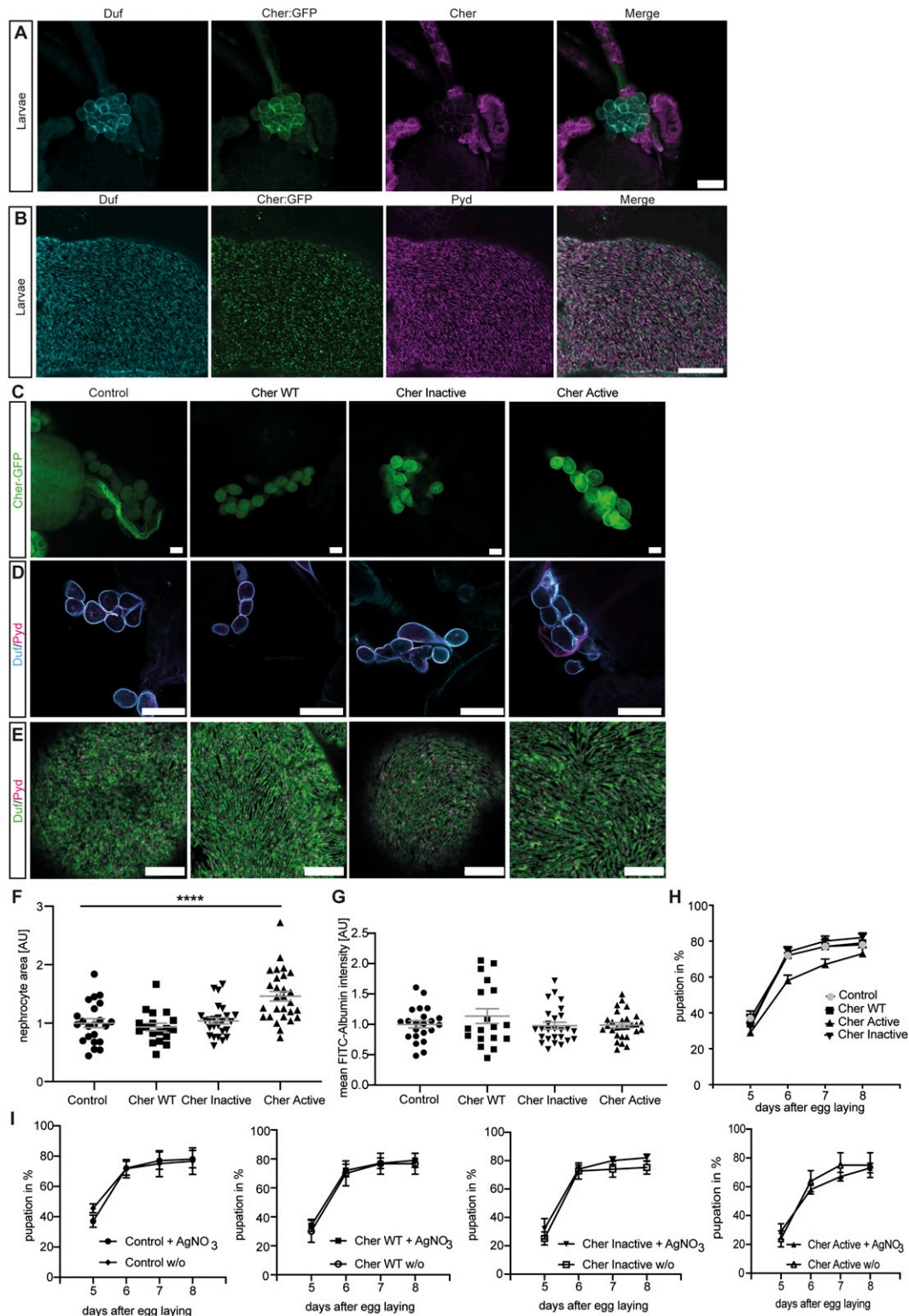

**Figure 1. Elevated Cheerio-MSR active levels result in nephrocyte hypertrophy.**

**(A)** Cheerio expression was confirmed in garland nephrocytes isolated from the third instar larvae (green: Cheerio:GFP combined with anti-GFP antibody; magenta: anti-Cheerio antibody). Co-staining with a Duf antibody (cyan) revealed partial co-localisation with Cheerio. Scale bar = 25 µm. **(B)** Confocal microscopy combined with Airyscan revealed co-localisation of endogenous Cheerio (green) with Duf (cyan). Pyd: magenta. Scale bar = 5 µm. **(C)** Cheerio mutants were expressed specifically in nephrocytes and are fused to a C-terminal GFP. Confocal microscopy revealed a cytoplasmic localisation of wild-type (WT) and inactive Cheerio (Cher), whereas the active variant shows an accumulation at the cell cortex. Scale bar = 25 µm. Control: w;*sns*-Gal4; UAS-*dicer2*, Cher wild type: w;*sns*-Gal4/UAS-*cher-wild type-MSR*;UAS-*dicer2*/+, Cher

additional proteins exhibit a mechanosensor function in podocytes remains unknown.

We and others have found the mechanosensor Filamin B to be up-regulated upon podocyte injury in mice and in patient-derived glomerular tissue (Koehler et al, 2020; Greiten et al, 2021; Okabe et al, 2021). It is not known if this up-regulation is a protective mechanism or induced as part of the pathological injury. Moreover, the functional role of Filamin B in podocytes remains largely unknown until today, as a global loss of Filamin causes embryonic (E14.5) or postnatal (directly after birth) lethality (Dalkilic et al, 2006; Feng et al, 2006; Hart et al, 2006; Zhou et al, 2010). Filamins are evolutionarily highly conserved, containing a N-terminal actin-binding domain and 24 $\beta$–sheet Ig domains, of which domain 16–24 contribute to the mechanosensor region (Razinia et al, 2012) (Fig S1A). Vertebrates have three Filamins (Filamin A, B, and C), whereas the *Drosophila* and *Caenorhabditis elegans* genomes encode a single Filamin (Cheerio and FLN-1, respectively) (Razinia et al, 2012).

Here, we used the *Drosophila* nephrocyte model to unravel the functional role of Filamin B and to specifically address whether Filamin B has a protective role in nephrocytes depleted in the key nephrocyte diaphragm proteins Sns and Duf.

# Results

## Cheerio is expressed in nephrocytes and co-localizes with the nephrocyte diaphragm protein Duf

Previous studies showed Filamin B levels to be elevated in patients suffering from focal segmental glomerulosclerosis (FSGS) and in injured murine podocytes, but its functional role and whether the increased expression levels are protective or pathogenic, remain elusive (Koehler et al, 2020; Okabe et al, 2021). Hence, within this study, we investigated the functional role of the *Drosophila* Filamin homologue, Cheerio, in more detail. To do so, we used the nephrocyte model, which shows a high morphological and functional similarity with mammalian podocytes and is widely used as a model for podocytes (Weavers et al, 2009; Hermle et al, 2017; Koehler et al, 2020).

We found that Cheerio is expressed from late embryonic stages and maintained throughout the life cycle in *Drosophila* garland nephrocytes (Figs 1A and B and S1A). Interestingly, Cheerio is found at the cortex of nephrocytes where it overlaps with the nephrocyte diaphragm proteins Dumbfounded (Duf/dNeph) and Pyd (dZO1), suggesting Cheerio is present at the nephrocyte diaphragm and might be part of the nephrocyte diaphragm multi-protein complex (Fig 1A and B). Based on its expression in nephrocytes and subcellular localisation to the nephrocyte diaphragm, we reasoned that the nephrocyte would be an excellent system to model Cheerio/Filamin B function.

## Elevated active Cheerio levels resulted in nephrocyte hypertrophy

$\beta$1-Integrin, a transmembrane protein that forms part of the extracellular matrix, is crucial for mechanotransduction. Both Filamin and Cheerio are able to bind $\beta$1-Integrin and the actin cytoskeleton. Filamin/Cheerio therefore provides a functional link between the extracellular matrix and the actin-cytoskeleton and allows extracellular force to be transduced into the cell (Razinia et al, 2012). To investigate the functional role of Cheerio and to mimic the elevated Filamin B levels observed upon podocyte injury, we assayed three different Cheerio variants in nephrocytes (Fig S1B). Here, we focused on the functional role of the mechanosensor region (MSR). In addition to the wild-type MSR (Cher wild type), we used a "closed" variant which requires greater mechanical force to unmask the binding sites for downstream targets of Cheerio, between domains 15/16 and 17/18, and is therefore less active (Cher Inactive) (Nakamura et al, 2006; Huelsmann et al, 2016). In addition, we used an "open" variant which requires less mechanical force to unmask the interaction sites of downstream targets and is therefore more active (Cher Active). The mutations introduced into Cheerio to generate the active variant were reported to cause an enhanced integrin binding (Lad et al, 2007). The interaction with $\beta$1-Integrin is mediated by a force induce conformational change of Filamin, which results in access to domain 21 (Filamin) (Sutherland-Smith, 2011).

To unravel the functional role of Cheerio-MSR, we expressed the Cheerio variants with a nephrocyte-specific driver (Sns-Gal4). We first analysed the subcellular expression pattern of each variant using the C-terminal GFP tag. Cher wild-type and Cher Inactive were localized throughout the nephrocyte cytoplasm. In contrast, Cher Active accumulated predominantly at the nephrocyte periphery, indicative of localisation at the nephrocyte diaphragm (Figs 1C and S1C). The discrepancy in the localisation between endogenously tagged Cheerio (Fig 1A) and over-expression of the wild-type Cheerio mechanosensor region (Fig 1C) seems to be a result of expression levels, as WT Cheerio-MSR is over-expressed using the UAS-Gal4 system and therefore protein levels exceed endogenous levels. Moreover, the Cheerio variants do not express the N-terminal actin-binding domain, which might also influence localisation in nephrocytes. This potential impact seems to be of minor importance, as the active variant, which is also lacking the actin-binding domain appears to be cortical.

Upon expression of the different Cheerio variants nephrocyte number and nephrocyte diaphragm integrity and morphology were not changed (Figs 1D and E and S7B). In contrast, nephrocyte size increased significantly when expressing Cher Active, but not Cher wild-type or Cher Inactive (Fig 1F). We also analysed nephrocyte function. The normal function of nephrocytes is to remove toxins

Inactive: w;*sns*-Gal4/+; UAS-*dicer2*/UAS-*cher-inactive-MSR*, Cher Active: w;*sns*-Gal4/+; UAS-*dicer2*/UAS-*cher-active-MSR*. **(D)** Duf and Pyd stainings did not show any changes in morphology after expressing the Cheerio variants. Scale bar = 25 $\mu$m. **(E)** Also, high-resolution imaging using Airyscan and confocal microscopy did not reveal any differences when compared with control cells (w;*sns*-Gal4; UAS-*dicer2*). Scale bar = 5 $\mu$m. **(F)** Measuring nephrocyte size resulted in a significant size increase in nephrocytes expressing the active Cheerio variant relative to control cells. One-way ANOVA plus Tukey's multiple comparisons test: ****$P$ < 0.0001. **(G)** FITC-Albumin uptake assays did not reveal any differences. **(H, I)** The AgNO$_3$ toxin assay showed a slightly delayed pupation in flies expressing the active Cheerio variant in nephrocytes; however, this difference was not significant.

from the haemolymph by filtration and endocytosis; processes which require an intact and functional nephrocyte diaphragm (Weavers et al, 2009). These processes can be assayed by (i) measuring FITC-Albumin uptake in isolated nephrocytes and (ii) monitoring the ability of nephrocytes to clear ingested $AgNO_3$ (a toxin) from the haemolymph in the intact animal (haemolymph borne $AgNO_3$ reduces viability and slows development, therefore haemolymph $AgNO_3$ concentration can be assessed indirectly by its impact on these parameters). We used both assays to determine the effects of Cheerio variants on nephrocyte function (Fig 1G–I). No defects were found for any of the variants in the FITC-Albumin uptake assay (Figs 1G and S1D). In line with this, we could not detect any significant delay in pupation behaviour for the different Cheerio variants in comparison with control flies (Fig 1H). When we compared the pupation behaviour of the different genotypes fed on $AgNO_3$ with animals fed on normal yeast we also could not detect any differences (Fig 1I). In summary, an increase in activated Cheerio stimulated hypertrophic growth of the nephrocyte but did not result in a pathological phenotype (i.e., there was no impact on nephrocyte morphology or filtration function). These data, along with the fact that hypertrophic growth is a protective mechanism during podocyte injury (Kriz & Lemley, 2015), suggest increased Cheerio/Filamin activity levels is a protective rather than pathogenic response.

### Cheerio exhibits a protective role in nephrocytes depleted in Sns/Duf

We next investigated whether Cheerio exhibits a similar protective role in abnormal nephrocytes. Loss of either of the nephrocyte diaphragm components Sns (dNephrin) or Duf resulted in a severe nephrocyte phenotype that includes both morphological and functional disturbances (Weavers et al, 2009; Zhuang et al, 2009). In detail, nephrocyte diaphragm integrity in Duf or Sns depleted cells was severely disrupted and cell size reduced, whereas FITC-Albumin uptake is significantly decreased and the onset of pupal development is severely delayed in the $AgNO_3$ toxin uptake assay. In line with our hypothesis that Cheerio provides a protective role, we found that endogenous Cheerio translocated from the cytoplasm to the nephrocyte cortex upon Sns or Duf depletion (Fig 2A, arrowheads).

To assess if Cheerio has a beneficial effect in abnormal nephrocytes, we induced a pathological phenotype (by depleting Sns or Duf), whereas simultaneously expressing the active or inactive Cheerio-MSR variants (Figs 2, S2, and S3). Filtration function was restored as shown with the FITC Albumin uptake assays (Figs 2D and S3B), but nephrocyte morphology and size could not be rescued by expressing the active Cheerio (Fig 2B and C). Active Cheerio also rescued the toxin uptake defects observed when depleting Duf (Fig 2E and F). Expression of inactive Cheerio only resulted in a rescue of the toxin uptake defects in nephrocytes lacking Duf (Fig S2A–E and S3A).

This phenotype could be explained by an alteration in endocytosis rather than filtration as morphology and nephrocyte diaphragm integrity are not restored based on our immunofluorescence data. Hence, to further delineate whether the filtration rescue is mediated via endocytosis, we performed Rab7 immunofluorescence

staining in nephrocytes expressing active or inactive Cheerio. Rab7 is described to be involved in the endocytic pathway in the transition from late endosome to lysosomes (Vanlandingham & Ceresa, 2009) and was shown to be one of the important Rab GTPases in nephrocytes (Fu et al, 2017). We did not observe a significant difference in Rab7-positive vesicles in the cells when expressing either Cheerio variants (Fig S3C). Moreover, we assessed Rab7-positive vesicles in Cheerio depleted nephrocytes as they have an increased FITC-Albumin uptake (Koehler et al, 2020). However, there was no significant difference in Rab7-positive vesicles upon Cheerio depletion (Fig S3D). Based on these findings we concluded that the filtration rescue caused by active Cheerio expression is not induced via Rab7-mediated endocytosis.

Interestingly, the inactive variant translocated to and accumulated at the cell cortex in nephrocytes depleted in Duf or Sns, suggesting even the inactive variant is activated in nephrocytes lacking Duf or Sns, strengthening the notion that Cheerio provides a protective function in abnormal nephrocytes (Fig S2A).

### Human Filamin B rescues cell size in abnormal nephrocytes

The wild-type human Filamin B (hFil B WT) displays a high sequence identity with Cheerio within the actin-binding domain (ACB, 68%) and MSR (52%) (Fig 3A). Hence, we generated flies expressing wild-type human Filamin B in nephrocytes. We first checked subcellular localisation of Filamin B in nephrocytes and find it localized close to the nephrocyte diaphragm, but it does not co-localize with the nephrocyte diaphragm protein Duf (Fig S4A and D). Interestingly, in contrast to Cheerio, expression of Filamin B did not result in hypertrophy, but caused a significantly increased FITC-Albumin uptake (Fig S4B and C).

To investigate whether the protective role we have found for Cheerio is conserved for Filamin B, we generated transgenic flies in which Sns or Duf were depleted while simultaneously expressing human Filamin B.

We found a significant rescue of the nephrocyte size by simultaneous expression of human Filamin B wild type, whereas morphology was partially restored (Fig 3B and C). The toxin uptake function in the $AgNO_3$ toxin assay was significantly rescued in Duf depleted cells although there was no rescue of the FITC Albumin uptake (Fig 3D–F), revealing some protective functions for Filamin B in abnormal nephrocytes.

To summarise the results for Cheerio and human Filamin B, the reduced cell size after Duf and Sns depletion could only be rescued by human Filamin B and not by Cheerio. Of note, the Cheerio constructs used do not possess the ACB, which could account for this functional difference. The functional assays revealed a significant rescue of the FITC-Albumin uptake phenotype for Cheerio and a significant rescue of the toxin uptake function for either Cheerio or human Filamin B under conditions where Duf was depleted. Neither Cheerio nor human Filamin B could restore morphology.

Taken together these data indicate that, similar to Cheerio, human Filamin B is protective in nephrocytes depleted in the diaphragm proteins Duf and Sns, although there are some differences in which phenotypes are rescued relative to Cheerio.

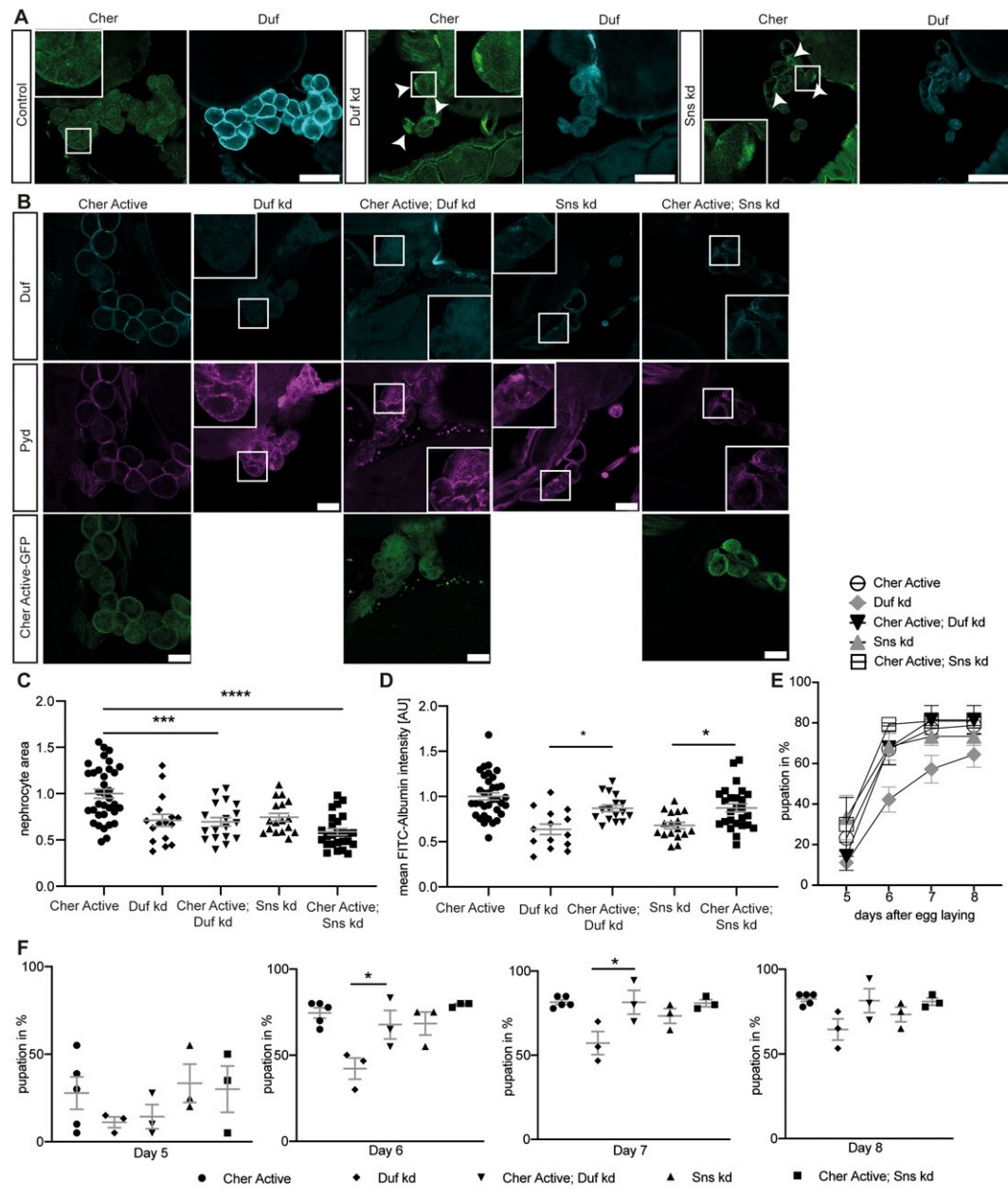

**Figure 2. Active Cheerio partially rescues the phenotype observed in abnormal nephrocytes.**
**(A)** Cheerio localisation was examined in abnormal nephrocytes. By depleting Duf and Sns, nephrocytes present with a severe morphological phenotype, as Duf expression and localisation is severely disrupted (cyan). Comparing Cheerio expression and localisation to control cells revealed an accumulation of Cheerio at the cell cortex in abnormal cells. Scale bar = 50 μm. **(B)** Active Cheerio was combined with Duf or Sns RNAi and showed no rescue of the nephrocyte morphology as depicted by Duf (cyan) and Pyd (magenta) staining. Scale bar = 25 μm. Cher active: w;sns-Gal4/+; UAS-cher-active-MSR/+, Duf kd: w;sns-Gal4/UAS-duf-RNAi; UAS-dicer2/+; Cher active; Duf kd: w;sns-Gal4/UAS-duf-RNAi; UAS-cher-active-MSR/+, Sns kd: w;sns-Gal4/UAS-sns-RNAi; UAS-dicer2/+; Cher active; Sns kd: w;sns-Gal4/UAS-sns-RNAi; UAS-cher-active-MSR/+.
**(C)** Nephrocyte size could not be rescued by simultaneous expression of active Cheerio and Duf or Sns RNAi. Cher Active was used as control and all other genotypes are depicted as relative values of Cher Active. One-way ANOVA plus Tukey's multiple comparisons test: ***$P < 0.001$; ****$P < 0.0001$. **(D)** FITC-Albumin uptake was restored in both, Duf- and Sns-depleted nephrocytes after expression of active Cheerio. One-way ANOVA plus Tukey's multiple comparisons test: ***$P < 0.001$. **(E, F)** The AgNO3 toxin assay revealed a rescue of the delayed pupation by expressing active Cheerio in combination with Duf RNAi. One-way ANOVA plus Tukey's multiple comparisons test: *$P < 0.05$.

## Hippo signalling induces nephrocyte hypertrophy and appears to be a downstream target of Cheerio

The above results show a hypertrophy phenotype and protective role for Cheerio, whereas human Filamin B restored cell size and filtration function. To identify the downstream pathway that mediates these effects, we performed a candidate screen examining three different pathways known to be involved in cell proliferation and tissue growth; TOR, Wingless (WNT) and Hippo (Huang et al, 2005; Laplante & Sabatini, 2012; Niehrs & Acebron, 2012). The over-activation of TOR, Yorkie (Yap/Taz; Hippo pathway) and armadillo (β-Catenin; WNT pathway) in nephrocytes resulted in a

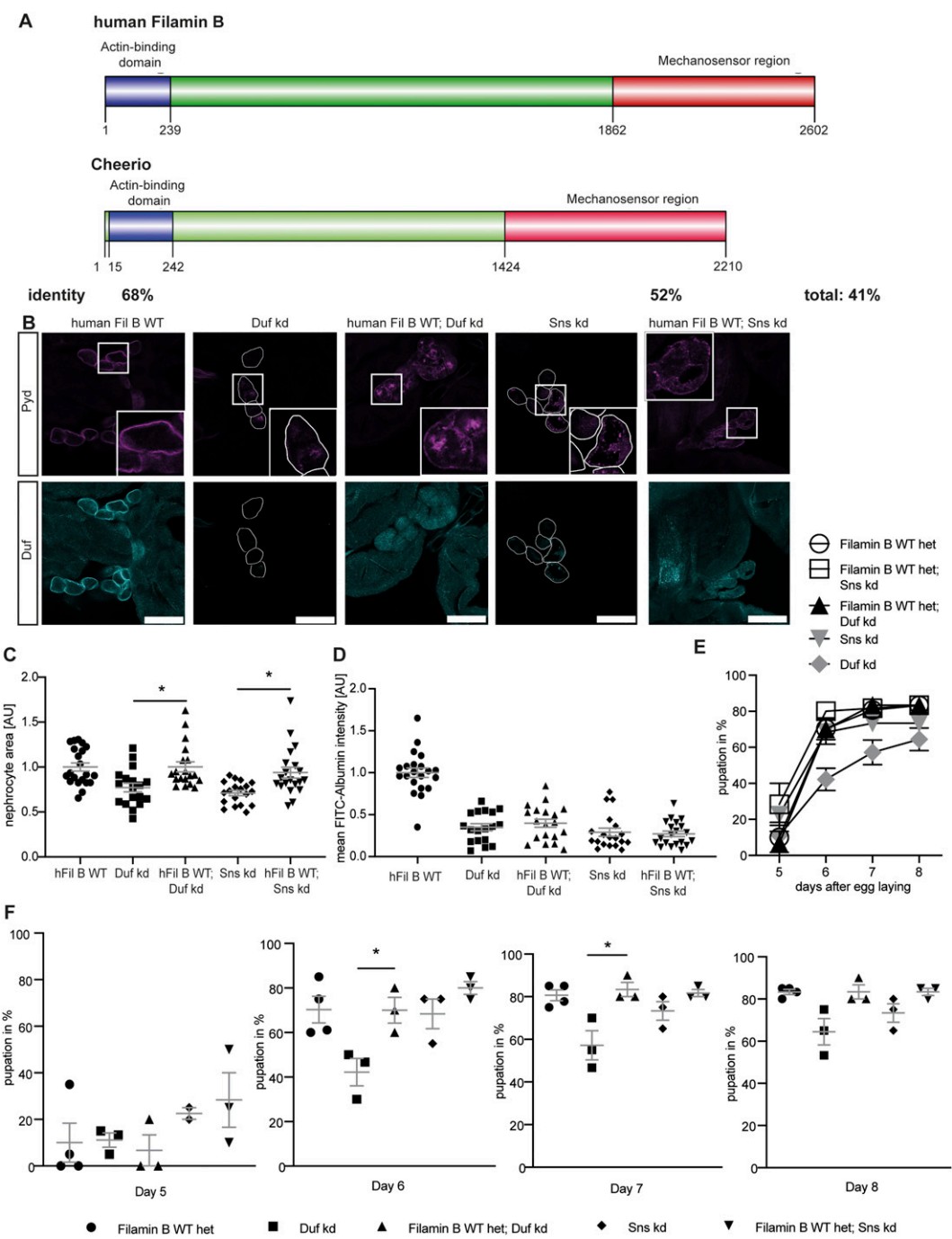

**Figure 3. Human Filamin B partially rescues nephrocyte size and function.**
**(A)** Schematic of human Filamin B wild type and *Drosophila* Cheerio wild type. Both possess an actin-binding domain at the N terminus, which shows a sequence identity of 68%. This domain is followed by the IgG domains, which form the mechanosensor region at the C terminus. The MSR shows a sequence identity of 52%. **(B)** Expression of human Filamin B wild type (human Fil B WT) in Duf and Sns depleted nephrocytes did partially restore morphology. Scale bar = 25 μm. hFil B WT: w;sns-Gal4/+; UAS-*hFilamin B WT*/+; Duf kd: w;sns-Gal4/+; UAS-*duf*-RNAi/UAS-*dicer2*; hFil B WT; Duf kd: w;sns-Gal4/+; UAS-*hFilamin B WT*/UAS-*duf*-RNAi; Sns kd: w;sns-Gal4/+; UAS-*sns*-RNAi/UAS-*dicer2*; hFil B WT; Sns kd: w;sns-Gal4/+; UAS-*hFilamin B WT*/UAS-*sns*-RNAi. **(C)** Nephrocyte size was restored by expressing human Filamin B wild type. One-way ANOVA plus Tukey's multiple comparisons test: *$P < 0.05$. **(D)** FITC Albumin uptake is not rescued by expression of the human Filamin B wild type. **(E, F)** AgNO$_3$ toxin assay revealed a rescue of the delayed pupation when human Filamin B wild type was expressed simultaneously to the Duf RNAi. One-way ANOVA plus Tukey's multiple comparisons test: *$P < 0.05$.

significant size increase to a degree similar to active Cheerio, consistent with the notion that one or more of these pathways acts downstream of Cheerio (Figs 4A–C and S5A–C). To help assess whether these pathways mediate the hypertrophic phenotype observed in nephrocytes expressing active Cheerio, we asked whether repression of these pathways is able to block the

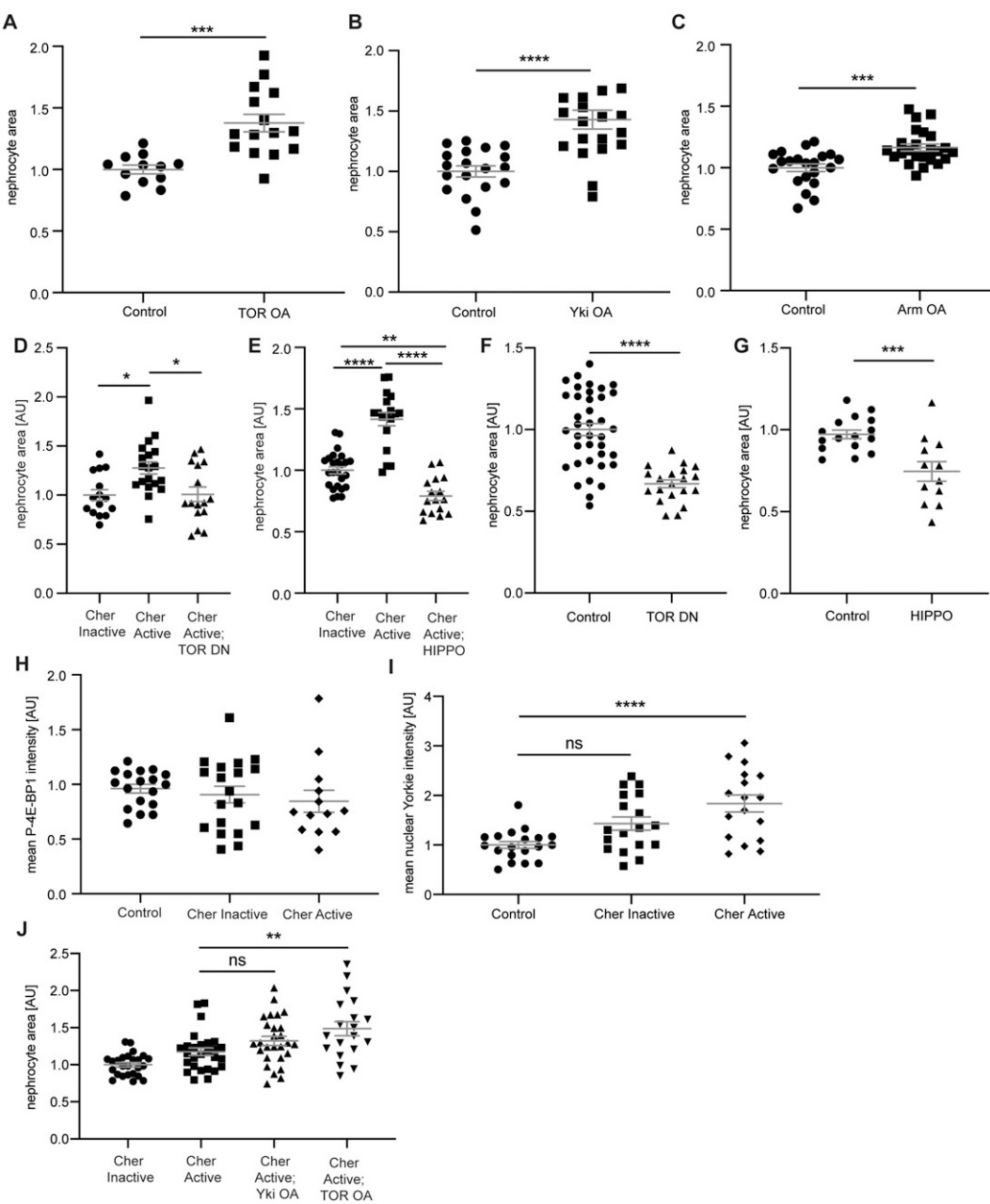

**Figure 4. Hippo/Yorkie signalling mediates the hypertrophy phenotype in nephrocytes.**
**(A)** Over-activation of TOR signalling in nephrocytes caused a significant hypertrophy phenotype. Control: w;sns-Gal4/+; UAS-*dicer2*/+; TOR OA: w;sns-Gal4/UAS-*rheb*;UAS-*dicer2*/UAS-*tor-wild type*. t test: ***P < 0.001. **(B)** Hyperactivation of Yorkie (Yap/Taz) in nephrocytes revealed a hypertrophy effect as well. Control: w;sns-Gal4/+; UAS-*dicer2*/+; Yki OA: w;sns-Gal4/+; UAS-*dicer2*/UAS-*yki.S111A.S168A.S250A.V5*. t test: ****P < 0.0001. **(C)** Expression of constitutively active Armadillo (β-Catenin) resulted in a significant size increase in nephrocytes. Control: w;sns-Gal4/+; UAS-*dicer2*/+; WNT OA: UAS-*arm.S10*/w;sns-Gal4/+; UAS-*dicer2*/+. t test: ***P < 0.001. **(D)** Quantification of the cell size based on GFP-tags to the Cheerio constructs revealed a significant size increase of Cher Active when compared with Cher Inactive cells. This increase was completely reversed when TOR signalling was inhibited with a dominant negative TOR variant (TOR DN). Cher Inactive: w;sns-Gal4/+; UAS-*cher Inactive*/+; Cher Active: w;sns-Gal4/+; UAS-*cher active*/+; Cher Active; TOR DN: w;sns-Gal4/+; UAS-*cher active*/UAS-*tor-dominant-negative*. One-way ANOVA plus Tukey's multiple comparisons test: *P < 0.05.
**(E)** Combination of Cher Active with Hippo, which results in the inhibition of Yorkie (Yap/Taz) caused a significant reduction of nephrocyte size, even below the cell size of Cher Inactive nephrocytes. Cher Inactive: w;sns-Gal4/+; UAS-*cher Inactive*/+; Cher Active: w;sns-Gal4/+; UAS-*cher active*/+; Cher Active; HIPPO: w;sns-Gal4/+; UAS-*cher active*/UAS-*hippo(dMST).flag*. One-way ANOVA plus Tukey's multiple comparisons test: **P < 0.01; ****P < 0.0001. **(F)** FITC Albumin assays revealed a significant size decrease when dominant negative TOR is expressed exclusively. Control: w;sns-Gal4/+; UAS-*dicer2*/+; TOR DN: w;sns-Gal4/+; UAS-*dicer2*/UAS-*tor-dominant-negative*. t test: ****P < 0.0001. **(G)** Expression of Hippo also resulted in a significant size decrease as assessed with FITC Albumin assays. Control: w;sns-Gal4/+; UAS-*dicer2*/+; HIPPO: w;sns-Gal4/+; UAS-*dicer2*/UAS-*hippo(dMST).flag*. t test: ***P < 0.001. **(H)** Quantification of the mean intensity of phosphorylated 4E-BP1 based on immunofluorescence staining did not reveal any significant differences in nephrocytes expressing active or inactive Cheerio in comparison with control cells. **(I)** Quantification of immunofluorescence staining using an antibody against Yorkie revealed a significantly higher mean nuclear intensity in nephrocytes expressing active Cheerio when compared with control cells. One-way ANOVA plus Tukey's multiple comparisons test: ****P < 0.0001. **(J)** Nephrocyte size was significantly increased when combining active Cheerio with over-active TOR, whereas the combination of active Cheerio with over-active Yorkie did not cause a significant size increase. One-way ANOVA plus Tukey's multiple comparisons test: **P < 0.01.

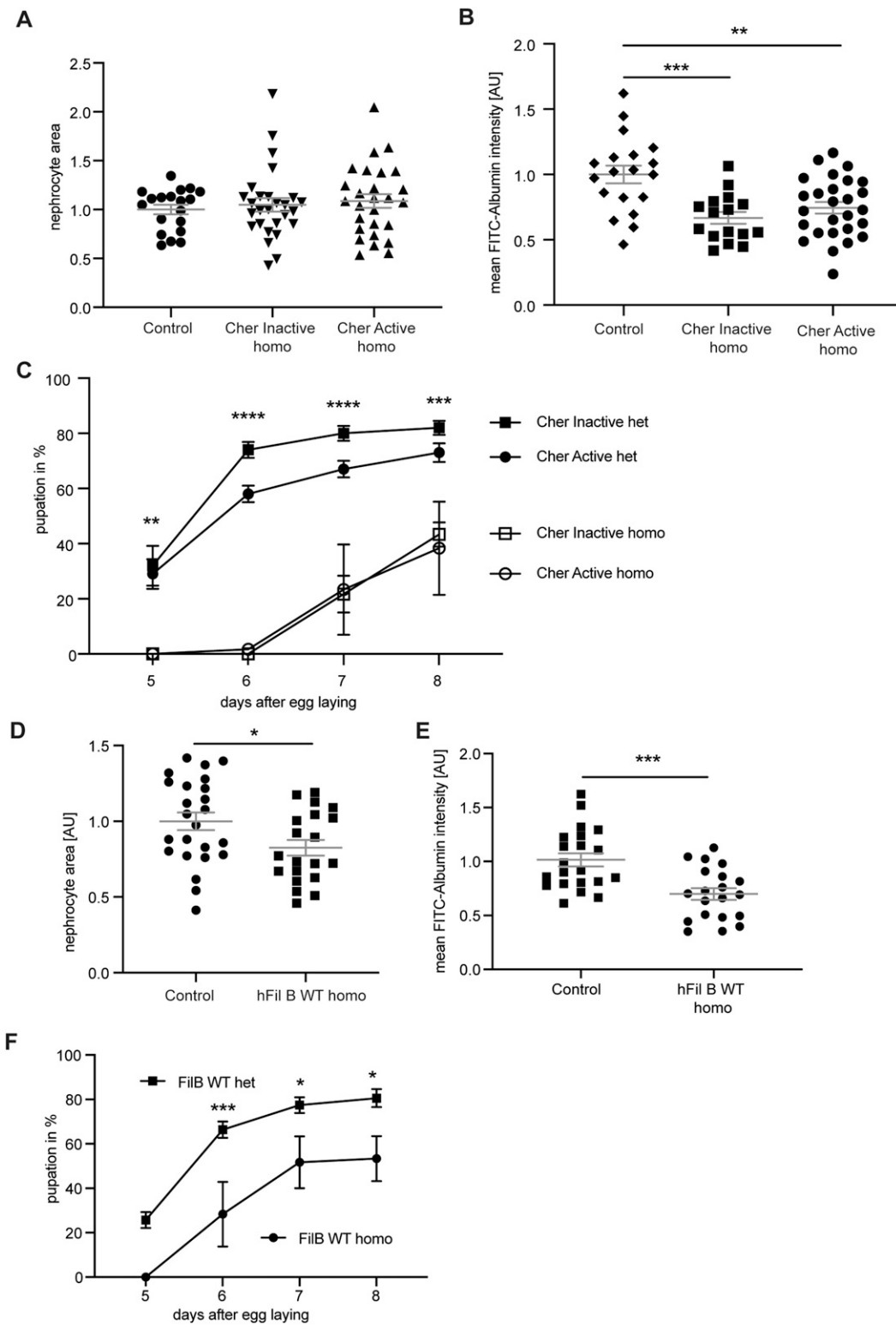

**Figure 5. Excessive increase of Cheerio and Filamin B levels results in a pathological response.**
**(A)** Homozygous Active or Inactive Cheerio were used to increase the level of protein expression in nephrocytes. Comparison with control flies (w;*sns*-Gal4; MKRS/TM6B) revealed no difference in nephrocyte size. Cher Inactive: w;*sns*-Gal4; UAS-*cher-inactive*; Cher Active: w;*sns*-Gal4; UAS-*cher-active*. **(B)** FITC-Albumin uptake was significantly decreased in both genotypes, inactive and active Cheerio. One-way ANOVA plus Tukey's multiple comparisons test: **$P < 0.01$; ***$P < 0.001$. **(C)** The AgNO$_3$ toxin assay revealed a severely delayed pupation in homozygous flies when compared with their heterozygous controls. Cher Inactive het: w;*sns*-Gal4/+; UAS-*cher-inactive*/UAS-*dicer2*; Cher Active: w;*sns*-Gal4/+; UAS-*cher-active*/UAS-*dicer2*; Cher Inactive homo: w;*sns*-Gal4; UAS-*cher-inactive*; Cher Active homo: w;*sns*-Gal4; UAS-*cher-active*.

hypertrophic phenotype caused by active Cheerio. In detail, we expressed a dominant-negative TOR variant (TOR-DN), or over-expressed Hippo (dMST) to repress Yki, in combination with expression of active Cheerio. (We also attempted to address whether WNT signalling was implicated by expressing a constitutive repressor version of the WNT effector pangolin [dTCF] [WNT DN]. We were unable to interpret this experiment as WNT pathway repression resulted in the complete loss of nephrocytes [Fig S5D], suggesting an important role of WNT signalling for the development and maintenance of nephrocytes, which is in line with previous reports for *Drosophila* and mammalian models [Carroll et al, 2005; Iglesias et al, 2007; Dai et al, 2009; Hurcombe et al, 2019].) Repression of TOR signalling blocks the active Cheerio-dependent size increase, restoring nephrocytes back to normal size (Cheerio inactive) (Figs 4D and S5E). Similarly, repression of Yki by expressing Hippo also blocks the active Cheerio-dependent size increase (Figs 4E and S5F). These data are compatible with TOR and Yorkie acting in a pathway(s) downstream of Cheerio. However, we also noted that TOR repression or Hippo over-expression in a wild-type background (i.e., without activated Cheerio), both resulted in a significant size decrease (Figs 4F and G and S5G and H). Therefore, we cannot exclude the possibility that TOR and Yorkie pathways act independently of Cheerio. Of note, our findings regarding cell size control mediated via TOR (activation and repression) are in line with recent findings by Spitz et al (2022).

To further investigate downstream signalling in Cheerio expressing nephrocytes, we performed immunofluorescence stainings to quantify TOR and Yorkie activation. In detail, we measured phosphorylation of 4E-BP1 using the phosphor-specific antibody p-4E-BP1 (TOR signalling leads to phosphorylation of 4E-BP1 and therefore provides a useful marker for TOR activation) and monitored Yorkie subcellular localisation (Yorkie translocates to the nucleus upon dephosphorylation). p-4E-BP1 levels in nephrocytes were equivalent to controls upon expression of active or inactive Cheerio (Figs 4H and S5I), indicating that the TOR pathway is not activated when manipulating Cheerio. In contrast, expression of active, but not inactive Cheerio resulted in a significant increase of nuclear Yorkie compared with control cells (Fig 4I and S5J). Thus, activation of Cheerio in nephrocytes inactivates the HIPPO pathway leading to hypertrophic growth. In addition, we performed genetic interaction studies combining active Cheerio with over-active Yorkie and TOR, to ask if hypertrophic growth is additive (evidence for a parallel pathway) or not (evidence for a common pathway). We find, active Cheerio in combination with over-active Yorkie did not cause a significant increase in hypertrophic growth compared with that of active Cheerio alone, providing additional evidence that Yorkie acts as a downstream target of Cheerio (Fig 4J). In contrast, active Cheerio combined with over-active TOR induced additional hypertrophic growth beyond that induced by active Cheerio suggesting that TOR and Cheerio act in parallel pathways (Fig 4J). Taken together, we provide data suggesting that Yorkie acts

downstream of Cheerio and mediates the hypertrophy phenotype observed in nephrocytes in response to activation of Cheerio.

## Excessive increase in Cheerio and Filamin B levels resulted in a pathological effect

Cheerio and Filamin B exhibit a protective role in nephrocytes depleted of Duf and Sns. Previous studies have shown that gain-of-function mutations in the mechanosensor TRPC6 caused a severe podocyte phenotype (Winn et al, 2005; Reiser et al, 2005). To address whether an excessive increase in Cheerio activity also results in a pathological phenotype, we generated lines homozygous for UAS-Cheerio-Active, UAS-Cheerio-Inactive or UAS-Filamin B (i.e., with two copies of the transgenes), with the aim of increasing the expression levels and therefore activity of Cheerio/Filamin B. Expression of a double dose of active or inactive Cheerio or human Filamin B produced a pathological phenotype, as nephrocyte morphology was severely disrupted, FITC-Albumin intensity was significantly decreased and toxin uptake function was significantly disrupted (Figs 5B, C, E, and F and S6A and C). Interestingly, while expression of homozygous human Filamin B resulted in a decreased nephrocyte size, the previously observed hypertrophy phenotype is not present in the homozygous Cheerio flies (Fig 5A and D). In further support of this, flies expressing the different Cheerio variants as single copy but raised at 28°C to increase protein levels, also presented with a severe morphological phenotype (Fig S6B). This suggests, together with the data described above, that a moderate increase in Cheerio activation is beneficial and protective in abnormal nephrocytes, but beyond a certain level Cheerio activation becomes pathological.

# Discussion

Recent studies revealed an important role of the mechanosensor Filamin in podocyte biology, as both Filamin A and B were described to be up-regulated under conditions of increased mechanical force, in several mammalian injury models and glomerular patient tissue (Koehler et al, 2020; Greiten et al, 2021; Okabe et al, 2021). In addition, it was previously shown that a nephrocyte-specific loss of Cheerio did not result in any change to nephrocyte diaphragm integrity or filtration function (Koehler et al, 2020). However, the function of Cheerio/Filamin in nephrocytes/podocytes is not known.

Based on these findings, we asked whether Cheerio exhibits a protective role in abnormal nephrocytes depleted in the key nephrocyte diaphragm proteins Duf and Sns. To investigate this hypothesis, we used the *Drosophila* nephrocyte model and examined the functional role of Cheerio, the Filamin homologue in the fly. Our data show that an over-activation of the mechanosensor domain caused a hypertrophy phenotype in nephrocytes, but did not impact on nephrocyte diaphragm integrity or filtration function. The link

---

Two-way ANOVA plus Tukey's multiple comparisons test: **$P < 0.01$; ***$P < 0.001$; ****$P < 0.0001$. **(D)** Homozygous expression of human Filamin B wild type resulted in a significant size decrease. *t* test: *$P < 0.05$. Control: Sns-Gal4; MKRS/TM6B: w;*sns*-Gal4; MKRS/TM6B; hFil B WT: w;*sns*-Gal4; UAS-*hFilamin B WT*. **(E)** FITC Albumin uptake was severely impaired by homozygous expression of human Filamin B WT when compared with controls. One-way ANOVA plus Tukey's multiple comparisons test: ***$P < 0.001$. **(F)** Comparison of homozygous and heterozygous controls in the AgNO$_3$ toxin assay also revealed severe filtration defects. Two-way ANOVA plus Tukey's multiple comparisons test: *$P < 0.05$; ***$P < 0.001$.

between Filamin B and the hypertrophy phenotype was also recently observed in a mouse model investigating secondary injured podocytes (Okabe et al, 2021). In a novel partial podocytectomy mouse model (loss of a subpopulation of podocytes) Okabe et al (2021) induced podocyte injury only in a subset of podocytes resulting in secondary damage in the remaining cells. Transcriptomic analysis revealed an up-regulation of Filamin B and a hypertrophy phenotype in the remaining podocytes (Okabe et al, 2021).

Of note, hypertrophy was previously described to serve as a protective mechanism during podocyte injury, as podocytes try to cover blank capillaries after loss of neighbouring cells by increasing their cell size (Wiggins et al, 2005; Puelles et al, 2019). As the Cheerio variants only express the mechanosensor region domain, the protective effects observed within this study might be linked to a mechano-sensitive mechanism.

Interestingly, Cheerio translocates and accumulates at the cell cortex in nephrocytes depleted of Duf and Sns. Previous studies in fibroblasts reported that active Filamin accumulates at the cell cortex, whereas inactive Filamin remains mainly cytoplasmic (Razinia et al, 2012; Nakamura et al, 2014). In line with this, we found that active Cheerio localized to the nephrocyte cortex, whereas the inactive variant remained cytoplasmic (Fig 6). Interestingly, inactive Cheerio (variant engineered to be less responsive to mechanical force) accumulates at the cortex in abnormal nephrocytes (upon loss of Duf and Sns). It is possible that in pathological conditions the forces experienced by Cheerio are such that even this "inactive" variant is activated and translocates to the cell periphery and nephrocyte diaphragm.

Our data also show that the expression of active Cheerio caused a rescue of nephrocyte function in Sns and Duf depleted cells (Fig 6). Similar effects were observed when expressing the inactive variant. Of note, inactive Cheerio seems to be activated in nephrocytes lacking Duf and Sns based on its localisation at the periphery. However, filtration function was rescued by active Cheerio, whereas nephrocyte morphology was not. This selective rescue could be explained by an alteration in endocytic capability as links between Filamin/Cheerio and endocytosis have been previously reported in literature with the suggestion that active Cheerio might mediate its protective role via endocytosis (Sverdlov et al, 2009; Lian et al, 2016; Jeong et al, 2017). We tested this by examining Rab7 (a marker for the transition from late endosomes to lysosomes) when expressing the Cheerio variants but did not observe a significant difference of Rab7-positive vesicles in nephrocytes expressing active or inactive Cheerio. Moreover, the overexpression of active Cheerio alone also did not impact on the FITC-Albumin uptake. We therefore concluded that the functional rescue of active Cheerio is not mediated via endocytosis, but rather filtration. Changes in nephrocyte diaphragm integrity sufficient to have an effect on filtration function might be too subtle to be resolved by confocal microscopy. Future studies including electron microscopy could help identifying these subtle changes. Also, active Cheerio might impact on the activity of the nephrocyte diaphragm complex, which serves as a signalling platform (Huber & Benzing, 2005; Weavers et al, 2009).

In addition, the protective effect mediated by Cheerio appears to be evolutionarily conserved in human Filamin B, although there are differences in the phenotypes which are rescued, suggesting that protein function does not completely overlap.

Moreover, our data provide evidence that the levels of Cheerio and Filamin B need to be tightly controlled as excessive levels result in a severe nephrocyte phenotype, indicating a threshold where protective effects tip toward a pathological condition (Fig 6).

We also provided evidence that hypertrophy is mediated by the Hippo pathway (Fig 6). Previously, several studies showed that loss of YAP in podocytes causes a glomerular disease (FSGS) phenotype (Schwartzman et al, 2016), that YAP activation has a protective effect during injury (Meliambro et al, 2017), and that nuclear localisation/activation of YAP has a pro-survival effect in podocytes (Bonse et al, 2018). Interestingly, in FSGS patient tissue inactive phospho-YAP levels were increased (Schwartzman et al, 2016; Meliambro et al, 2017). These findings, together with the up-regulated Filamin B levels in podocyte models, indicate that the mechanisms investigated within this project might be highly evolutionarily conserved and that YAP might be a downstream target of Filamin in mammals as well. Hippo signalling has been previously investigated in nephrocytes in the context of WNT signalling. Depletion of shaggy (dGSK3β) caused a nephrocyte phenotype, which is mediated via dysregulated Hippo signalling (Hurcombe et al, 2019). Moreover, blocking YAP/TAZ by Verteporfin treatment (causing activation of Hippo) results in a protective effect in nephrocytes in injury models (Hurcombe et al, 2019).

Here, we could show that Yorkie/YAP acts as downstream target of Cheerio and activating Hippo reversed the hypertrophy phenotype in nephrocytes expressing active Cheerio. Taken together,

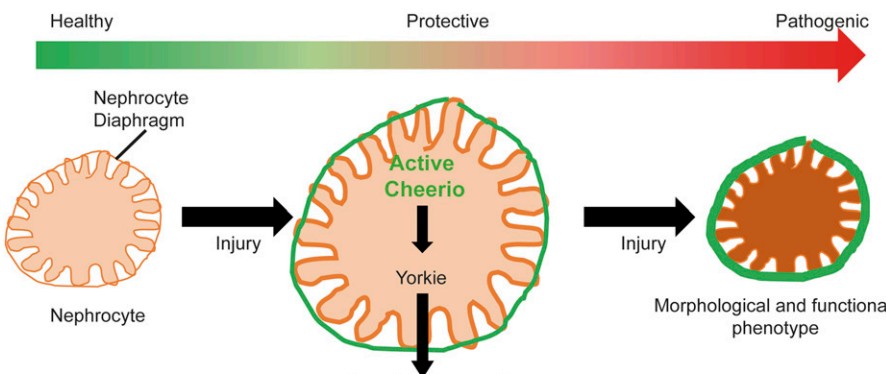

**Figure 6. The protective role of Cheerio in *Drosophila* nephrocytes.**
Cheerio is active in nephrocytes depleted of the nephrocyte diaphragm proteins Duf and Sns, and accumulates at the cell periphery. This activation and the increased protein levels result in a protective effect via hypertrophic growth, mediated via Hippo/Yorkie signalling. However, Cheerio levels need to be tightly controlled, as excessive increase of protein levels results in a shift from the protective to a pathogenic effect, including a morphological and functional phenotype.

both studies provide data that Hippo signalling is important for nephrocyte biology.

In a recent study, Greiten et al (2021) described the functional role of Filamin A in podocytes, and showed that loss of Filamin A caused a rearrangement of the actin-cytoskeleton and decreased expression levels of focal adhesion associated proteins (Greiten et al, 2021). They also showed increased Filamin A levels during hypertension in mouse models and hypertensive glomerular patient tissue (Greiten et al, 2021), again in line with our hypothesis for a protective role of Cheerio/Filamins when up-regulated at moderate levels. Filamin A and B are ~70% identical in amino acid sequence and therefore are likely to exhibit shared functions (Takafuta et al, 1998; Sheen, 2002). However, because of differences in the hinge regions Filamin B also seems to have unique functions, in particular in regard to membrane–cytoskeletal interactions (Takafuta et al, 1998).

Of note, Filamin B is not only expressed in podocytes, but also in mammalian proximal tubular cells, which raises the question as to what extent our identified and described mechanism is relevant for podocyte biology. In our study we were able to rescue a phenotype induced by loss of nephrocyte diaphragm proteins and Rab7-mediated endocytosis was not affected by overexpression or depletion of Cheerio. Moreover, it is not clear to what extend nephrocytes resemble proximal tubule function. Nephrocytes express cubilin and amnionless, which are also expressed in mammalian proximal tubular cells and are required for reabsorption processes (Zhang et al, 2013). This finding suggests nephrocytes are not only a model for podocyte biology, but also for proximal tubular cells. However, in recent studies cubilin and amnionless have been identified to be expressed in mammalian podocytes and to be important for nephrocyte diaphragm

remodelling in nephrocytes (Prabakaran et al, 2012; Gianesello et al, 2017; Atienza-Manuel et al, 2021). Moreover, according to single nucleus RNA-seq in adult fly kidneys, nephrocytes do not show an overlap with murine proximal tubular cells, but with podocytes and parietal epithelium (Xu et al, 2021 Preprint). The recent data and several studies using nephrocytes support the concept of nephrocytes as podocyte equivalent, but similarities to proximal tubular cells cannot be excluded.

Taken together, our data show that Cheerio is activated in abnormal nephrocytes upon loss of the nephrocyte diaphragm proteins Duf and Sns, which results in a protective hypertrophy phenotype mediated via Hippo signalling (Fig 6). This protective role for Cheerio and Filamin need to be tightly controlled; however, as an excessive increase in activity results in a shift from a protective to a pathological response (Fig 6).

# Materials and Methods

### Fly husbandry and generation

All flies were kept at 25°C for experiments (Table 1). The nephrocyte-specific expression was achieved by mating UAS fly strains to the Sns-Gal4 strain.

Human Filamin B isoforms were amplified from a human embryonic kidney cDNA library using the following primers:

hFilamin B WT (full length):

fwd: 5′-CGCGGGACGCGTACCATGCCGGTAACCG-3′
rev: 5′-ATGCACGCGGCCGCTTAAGGCACTGTGAC-3′

hFilamin B was cloned into the pUAST vector with a C-terminal HA tag.

**Table 1.  List of fly strains.**

| Fly strain | Origin | Purpose | Chromosome |
|---|---|---|---|
| Cher-GFP-Trap | BDSC ID: 60261 | Expression pattern | 3. Chr. |
| UAS-Cher-WT-MSR-GFP | Kindly provided by J Ylänne | Over-expression | 3. Chr. |
| UAS-Cher-inactive-MSR-GFP (Cher closed) | Kindly provided by J Ylänne | Over-expression | 3. Chr. |
| UAS-Cher-active-MSR-GFP (Cher open) | Kindly provided by J Ylänne | Over-expression | 3. Chr. |
| UAS-Sns-RNAi | VDRC ID: 109442 | Knockdown | 2. Chr. |
| UAS-Duf-RNAi | VDRC ID: 27227 | Knockdown | 2. Chr. |
| Sns-Gal4; UAS-Dicer2 | Selfmade | Nephrocyte-specific driver | 2. + 3. Chr. |
| UAS-human Filamin B WT | Selfmade/transgenics made by Genetivision | Over-expression | 3. Chr. |
| UAS-TOR-WT | BDSC ID: 7012 | Over-expression | 3. Chr. |
| UAS-TOR-DN | BDSC ID: 7013 | Over-expression | 2. Chr. |
| UAS-Rheb; UAS-TOR-WT (TOR-OA) | BDSC ID: 80932 | Over-expression | 2. + 3. Chr. |
| UAS-arm.S10 (constitutively active β-catenin) | BDSC ID:4782 | Over-expression | X Chr. |
| UAS-pan.dTCFdeltaN (WNT repressor) | BDSC ID:4784 | Over-expression | 2. Chr. |
| UAS-yki.S111A.S168A.S250A.V5 (YAP hyperactive) | BDSC ID:28817 | Over-expression | 3. Chr. |
| UAS-Hippo(dMST).Flag | BDSC ID:44254 | Over-expression | 3. Chr. |
| Oregon R | | Wild type | |
| W[1118] | | Wild type | |

## Immunofluorescences of *Drosophila* tissue

*Drosophila* embryos were collected over 24 h and dechorionated in 50% bleach for 10 min. After a washing step with $H_2O$, they were fixed in 4% Formaldehyde/Heptane for 20 min. Heptane was then removed and replaced by Methanol, vortexed for 30 s to devitellinized. The two phases separated after a minute and the embryos remained in the lower formaldehyde phase. Both phases were removed and the embryos were washed with methanol x3 for 20 min. The embryos were then washed with washing buffer (phosphate-buffered saline with 0.5% BSA and 0.3% Triton-X, PBS-Tx-BSA) x3 for 20 min, followed by overnight incubation at 4°C in primary antibody (see Table 2). Primary antibody was removed from embryos, and nonspecific antibody binding was removed with 3× washes in PBS-Tx-BSA, the embryos were blocked in PBS-Tx-BSA + 5% normal horse serum for 30 min, and then incubated with the appropriate secondary antibody for 1 h at room temperature (see Table 2). Embryos were washed 3 × 10 min in PBS-Tx-BSA before mounting in mounting medium (2.5% propyl gallate and 85% glycerol).

Garland nephrocytes were prepared by dissecting a preparation containing the oesophagus and proventriculus (to which the garland nephrocytes are attached) from third instar larvae in haemolymph-like buffer (HL3.1; 70 mM NaCl, 5 mM KCl, 1.5 mM $CaCl_2$, 4 mM $MgCl_2$, 10 mM $NaHCO_3$, 115 mM sucrose, and 5 mM Hepes). The cells were fixed in 4% formaldehyde for 20 min, followed by three washing steps with PBS-Tx-BSA for 20 min. Incubation in primary antibody was at 4°C overnight. The next day, the preparation was washed x3 for 20 min, blocked with 5% normal horse serum (in PBS-Tx-BSA) for 30 min, and followed by incubation in the appropriate secondary antibody for 1 h at room temperature. The preparation was washed x3 for 20 min, before mounting in mounting medium. Imaging was carried out using either a Zeiss LSM 800 confocal or a Leica SP8 confocal. Images were further processed using ImageJ (version 1.53c).

## FITC-albumin assay

FITC-Albumin uptake assays were performed as previously published (Hermle et al, 2017; Koehler et al, 2020). Garland nephrocytes were prepared as described above and incubated with FITC-Albumin (0.2 mg/ml; Sigma-Aldrich) for 1 min, followed by 20-min fixation with 4% formaldehyde. Nephrocytes isolated from Cheerio over-expression flies were used for additional antibody staining against Albumin (Cy3-labelled secondary antibody), as they express GFP. Antibody-based staining was performed as previously described. For comparative analysis, the exposure time of experiments was kept identical. The mean intensity of FITC-Albumin or the Cy3-labelled FITC-Albumin was measured and normalized to control cells. Images were also used to quantify nephrocyte size by measuring the area of each cell. As experimental approaches were different for some experiments used for size quantification, we show quantified values in the main figures for better comparison, but all absolute values of nephrocyte size are depicted in Fig S7.

## AgNO$_3$ toxin assay

Flies of the appropriate genotype were allowed to lay eggs for 24 h at 25°C on juice plates with yeast. Plates with eggs were incubated at 18°C for 24 h. A defined number of first instar larvae (usually 20) were then transferred to juice plates supplemented with yeast paste containing $AgNO_3$ (2 g yeast in 3.5 ml 0.003% $AgNO_3$). Plates were kept at 25°C for 4 d to let larvae develop and pupate. The number of pupae was counted daily between days 5 and 9 after egg laying and percentage pupation (relative to the number of larvae added to plate) was calculated.

## Statistical analysis

One data point represents one fly and intensity as well as size was measured for every nephrocyte and used for calculation of mean values per fly. To determine statistical significance Graph Pad Prism software version 8 for Mac (GraphPad Software) was used. All results are expressed as means ± SEM. Comparison of more than two groups with one independent variable was carried out using one-way ANOVA followed by Tukey's multiple comparisons test. A *P*-value < 0.05 was considered statistically significant.

Table 2.  List of antibodies.

| Name | Company/provider | Catalog no./reference | Host species | Dilution IF |
|---|---|---|---|---|
| Albumin | Abcam | ab207327 | Rabbit | 1:25 |
| Cher | M. Uhlirova | Külshammer and Uhlirova (2013) | Rat | 1:100 |
| Duf | M. Ruiz-Gomez | Weavers et al (2009) | Rabbit | 1:100 |
| GFP | Abcam | ab6673 | Goat | 1:200 |
| HA tag | Proteintech | 51064-2-AP | Rabbit | 1:100 |
| HA tag (for hFilB wt IF) | Cell Signalling | 2367S | Mouse | 1:100 |
| Phospho-4E-BP1 | Cell Signalling | 9459S | Rabbit | 1:100 |
| Pyd | Developmental Studies Hybridoma Bank | PYD2 | Mouse | 1:25 |
| Rab7 | Developmental Studies Hybridoma Bank | Rab7 | Mouse | 1:25 |
| Yorkie | K.D. Irvine | Oh and Irvine (2009) | Rabbit | 1:100 |

## Supplementary Information

## Acknowledgements

The Anti-Pyd monoclonal antibody developed by Fanning (Choi et al, 2011) was obtained from the Developmental Studies Hybridoma Bank, created by the NICHD of the NIH and maintained at the University of Iowa, Department of Biology, Iowa City, IA 52242. S Koehler received funding from the German Research Foundation (KO 6045/1). TB Huber received funding from the German Research Foundation (HU1016/11-1).

### Author Contributions

S Koehler: conceptualization, data curation, formal analysis, funding acquisition, validation, investigation, visualization, methodology, project administration, and writing—original draft, review, and editing.
TB Huber: resources, funding acquisition, and writing—review and editing.
B Denholm: conceptualization, resources, funding acquisition, and writing—review and editing.

### Conflict of Interest Statement

The authors declare that they have no conflict of interest.

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
