## [Reviewer comments · Life Science Alliance]

Life Science Alliance

A protective role for *Drosophila* Filamin in nephrocytes via Yorkie mediated hypertrophy

Sybille Koehler, Tobias Huber, and Barry Denholm

DOI: <https://doi.org/10.26508/lsa.202101281>

Corresponding author(s): Sybille Koehler, Universität Hamburg

Review Timeline:

Submission Date:	2021-12-02
Editorial Decision:	2021-12-12
Revision Received:	2022-05-25
Editorial Decision:	2022-06-17
Revision Received:	2022-07-13
Editorial Decision:	2022-07-14
Revision Received:	2022-07-15
Accepted:	2022-07-18

Scientific Editor: Novella Guidi

Transaction Report:

December 12, 2021

Re: Life Science Alliance manuscript #LSA-2021-01281-T

Dr. Sybille Koehler
University of Edinburgh
George Square
Edinburgh, Lothian EH8 9XF
United Kingdom

Dear Dr. Koehler,

Thank you for submitting your manuscript entitled "Drosophila Filamin exhibits a mechano-protective role during nephrocyte injury via hypertrophy" to Life Science Alliance. The manuscript was assessed by expert reviewers, whose comments are appended to this letter. As you will note from the reviewers' comments below, both the reviewers are interested about the study but do raise some important concerns that need to be addressed in the revision, with the common one being the presence of overstated conclusions in which there is a lack of experimental data to support mechanotransduction and mechanoprotective role on nephrocyte function or phenotype. Please address also all the issues related to figures presentation and quantification. We, thus, encourage you to submit a revised version of the manuscript back to LSA that responds to all of the reviewers' points.

Thank you for this interesting contribution to Life Science Alliance. We are looking forward to receiving your revised manuscript.

Sincerely,

B. MANUSCRIPT ORGANIZATION AND FORMATTING:

Reviewer #1 (Comments to the Authors (Required)):

The manuscript by Koehler and Denholm uses *Drosophila* nephrocytes to test the hypothesis that the Integrin-actin crosslinking protein Cheerio/Filamin serves a protective role during injury. Using genetic variants affecting Cheerio's ability to interact with Integrin, the authors find that activated Cheerio leads to cellular hypertrophy, which they go on to suggest is mediated by TOR and Hippo signaling. In the context of nephrocyte injury (involving disruption of slit diaphragms and filtration function), Cheerio can partially rescue nephrocyte filtration and a related delay in pupation. They also show some ability of human FilaminB to improve certain aspects of the nephrocyte injury model. The experiments are well-conducted, including quantification of multiple phenotypes across multiple genotypes. The study presents several novel findings that will be of interest to the field, and is therefore appropriate for publication in Life Science Alliance, however some of the conclusions seem overstated based on the current data. As discussed below, either additional experimental support should be provided, or the writing should be modified to reflect that some of the data are simply consistent with the author's interpretation/model, but are not themselves conclusive evidence.

Major comments

Cheerio localization:

In Figure 1 and associated text, the authors conclude that Cheerio partially colocalizes with the slit diaphragm protein Duf, which they suggest may indicate Cheerio is in complex with slit diaphragm-associated proteins. However, the images provided (Figure 1A,B) are very low magnification, which makes determining its precise localization difficult. Can the localization of Cheerio be determined in high-resolution images similar to those in panel E? This would be particularly useful in examining the localization of the wildtype, inactive, and active forms of the protein. Similar high magnification images would be helpful for Human FilB, which they suggest also localizes to slit diaphragms (Supplemental Figure 4A). If these types of images cannot be provided, it seems necessary to significantly tone down the conclusion that these proteins are located at slit diaphragms.

The methods section (Table 2) suggests the three Cheerio MSR variants from the Huelsmann 2016 paper are UAS, however the description in that paper indicates they are knock ins at the endogenous locus (attB-mediated replacement of the 3' end). If this is correct, it will change the wording in places. For example, the text indicates these constructs are being driven by *Sns-Gal4*, but instead they would be expressed from the endogenous cheerio promoter. It may also be more accurate to indicate that expression of the Cheerio-MSR Active form of the protein causes hypertrophy, as opposed to "elevated" levels of Cheerio-MSR Active protein.

There are notable differences in the localization patterns of Cheerio:GFP and the anti-Cheerio staining, particularly in panel A. Is there an explanation for this? They appear more similar (cortical) in panel B.

I am also confused by the wildtype Cheerio:GFP localization patterns in panel B and C. I believe the Cheerio:GFP in B is the GFP trap line mentioned in the methods section (this was not clear in the figure legend). Is there an explanation for why the wildtype Cheerio:GFP in panel B appears cortical while the Cheerio:GFP with wildtype MSR (essentially the wildtype protein I believe) is cytoplasmic?

Cheerio exerts a protective effect in damaged nephrocytes:

Figure 2. Panel D suggests that Cheerio-Active can rescue FITC:Albumin uptake caused by knockdown of slit diaphragm proteins. Knockdown of these proteins disrupts the pattern of slit diaphragms on the nephrocyte surface, and presumably reduces uptake of fluorescent tracers due to subsequent loss of labyrinthine channel area, where the majority of endocytic uptake of these tracers occurs. However, the current data (Figure 2B) suggest that slit diaphragms are not restored by Cheerio-Active. Is Cheerio-Active directly affecting rates of endocytosis through some other mechanism? It would be helpful if the authors could provide more discussion of possible mechanisms underlying these findings.

In their previous study (Koehler et al., JASN 2020), Cheerio knockdown had no apparent effect on otherwise normal nephrocytes. The model proposed here suggests that Cheerio has a protective effect in injured podocytes/nephrocytes. Thus, if Cheerio protects nephrocytes in response to injury, one would predict that knockdown of Cheerio in a damaged nephrocyte should significantly dampen that protective response, leading to stronger defects, be it cell morphology, size, or function. If the authors agree with this rationale, can they perform a double knockdown of Cheerio and *Sns* or *Duf* to assess whether this is

correct?

TOR and Yki are downstream of Cheerio:

Page 21, line 16. The subsection title of "TOR and Yorkie signaling induce nephrocyte hypertrophy and appear to be downstream targets of Cheerio" seems an overly strong conclusion because, as the authors note in the manuscript, the expression of TOR DN or HIPPO alone decrease cell size, thus the apparent rescue of Cher Active hypertrophy may simply be the averaged effects of two independent processes acting on cell size. It may support their conclusion if they could show increased activity for these pathways in Cher Active cells. There are available genetic reporters for Hippo activity (Parker and Struhl, PLOS Biology 2015). TOR activity has been assessed in flies using various readouts, such as phospho-4E-BP antibody staining (Teleman et al., Developmental Cell. 2005). Similarly, based on the current data, the proposed relationship between Cheerio and TOR/Yki in Figure 6 should be made less conclusive; perhaps by making the arrows dashed or adding question marks. The figure legend text should also reflect that these relationships are somewhat tentative.

Minor comments

Figure 1 Is titled: "Elevated Cheerio-MSR levels result in nephrocyte hypertrophy", however, because the hypertrophy only results from the active form, perhaps more specific language could be used, such as "Elevated levels of Cheerio-MSR-Active result in nephrocyte hypertrophy".

Figure 2 The order of genotypes in the images in panel B is different from the x-axis of graphs in panels C/D. Similarly, the order of genotypes within panels C/D is different from the x-axis in F. Since these include all of the same genotypes, it would be helpful to the reader if the order of genotypes could be kept consistent. The order of genotypes in these graphs is also different from that in Supplemental Figure 2. This makes it confusing when trying to compare the effects of active vs inactive Cheerio in the different genotypes.

It is not possible to distinguish the genotypes by symbols in Figure2 panel D. Can the symbols and lines also be color coded or the symbols enlarged or otherwise made more obvious?

The figure legend for panels E,F state that Cheerio Active rescues delayed pupation of Duf or Sns RNAi, but the data only indicate a delayed pupation for Duf (and associated rescue). Thus it seems reference to a rescue of Sns RNAi should be omitted here.

Figure 3 In Panel B, it appears that expressing hFilB in Duf kd cells is somehow restoring Duf expression. Is this correct? If so, can the authors provide any explanation?

In Panel D, there is a "*****" but no line indicating which genotypes are statistically different. The figure legend suggests none of them are different, so the "*****" probably just needs to be removed.

Figure 4 Can the data in panels F and G be merged together with D and E, respectively, to make comparisons between all genotypes more straightforward?

Page 19, line 8 indicates that the Cheerio constructs do not possess the ACB, but as discussed above, I do not believe this is accurate. The mutant Cheerio knock-ins described in the Huelsmann 2016 paper suggest the MSR variants should be part of the full-length protein (long isoform), which should therefore contain the ACB. I believe this also affects the interpretation of the data as presented on page 29, line 10, where the inference is made that hFilB rescues cells size due to possessing the ACB, thus the ACB is important for cell size, however if it is correct that the Cheerio variants are indeed full-length proteins possessing the ACB, this conclusion does not seem supported.

Yki is the common abbreviation for Yorkie, not "Yrk".

Page 24, the text does not make it clear that the phenotypes result from homozygosity for the Cheerio MSR Active or Inactive variants, not wildtype Cheerio.

Table 2 lists UAS-hFilB constructs delta ACB and delta MSR, but I do not believe there are any data or discussion of these lines in the manuscript. Probably remove these from table, along with associated primer information.

Reviewer #2 (Comments to the Authors (Required)):

The paper describes the effects of modulating the fly ortholog of human filamins. This is done in fly nephrocytes, which are used to study aspects of human kidney podocytes and proximal tubule cells. The paper focuses on nephrocytes as a model of podocytes but uses endocytic function as a marker, a model of proximal tubule cell function. Nephrocyte are both a model of both human podocyte and proximal tubules cell and this should really be mentioned. Also, the localisation of filamin B in human proximal tubules cells might suggest the drosophila model is better suited to establishing filamin's role in tubules cell endocytosis - given that its at the cell surface. Its not so simple to say that Drosophila filamin (Cher) models only filamin biology in human

podocytes. Apologies in advance if I've misunderstood or missed something I should have seen or considered.

1. The paper deals with Filamin's 'mechano-protective role' in nephrocytes however there needs to be experimental data presented in the paper to support this mechanotransduction role. At present there is no experimental data relating mechanotransduction, the mechano-protective role is inferred from other studies and not demonstrated directly.
2. Neither the introduction nor the discussion mentions a considerably important paper about YAP/TAZ signaling in mammalian podocytes and *Drosophila* nephrocytes by Hurcombe et al Nature Communications 2019. That paper's findings need to be mentioned and the current data examined within the context of Hurcombe's work.
3. Nephrocyte numbers should be stated for all experimental genotypes. If modulations are protective then nephrocyte numbers should be similar - this needs to be shown.
4. Nephrocyte area need to be stated in square microns, not as relative values.
5. Figure 1 and other images. The images of nephrocytes are very small, most of the image is extraneous area and the images do little to support the quantified data. From the images, the nephrocytes do not appear to be larger in the Cher active experimental group.
6. There is a large discrepancy between the albumin binding as quantified in figure 3 compared to albumin binding for similar genotypes in Figure 2 (70-80% reduced in figure 3 vs 30% reduced in Figure 2 for both the sns KD and duf KD flies).
7. Figure 3B the images should be presented in the same order as the data in the graphs (Fig 3C and D).
8. Figure 3D the asterisks needs an accompanying line.
9. Figure 3. Cells really need to be counterstained with some general cell marker. They cannot be seen in the Duf KD and other images.
10. Figure 3's figure legend: this is a little confusing me: the title states 'Human Filamin B rescues nephrocyte size and function', then for 3B it states 'Expression of human Filamin B wildtype (human Fil B WT) in Duf and Sns depleted nephrocytes did not restore morphology', then 3C states, 'Nephrocyte size was restored by expressing human Filamin B wildtype, then 3D states 'FITC Albumin uptake is not rescued by expression of the human Filamin B wildtype. If I've understood this, it looks like Human Filamin B rescues the AgNO₃ toxicity independently of rescuing albumin binding / endocytosis; which is odd. Yet function is stated as being restored - which is only partly the case.
11. Supp Fig 4 -colocalization data is required to support the claim that Human filamin B localises to the slit diaphragm. Currently it shows the HA tag and Pyd are at the cell surface but that's far from showing colocalization at the slit diaphragm. If the same logic was applied more generally, all cell surface proteins could be claimed to localise to the slit diaphragm, and that is not the case.
12. Figure 5. The control appears to comprise a line with two chromosome 3 balancers (MKRS/Tm6B). That's a lot of mutations that cannot be regarded as representative of wild type and therefore another control really needs to be used.
13. Discussion: Filamin B is expressed at the apical surface of human proximal tubule cells - cells modelled in the nephrocyte via the endocytosis assays. Proximal tubule cells are never mentioned in the paper and that needs addressed, the nephrocyte is not just a model of podocytes. It is unclear who these findings from the fly model relate to podocytes alone.
14. There is persistent reference to 'cell injury' yet there is no insult to the cells that constitutes an injury as such; injury is due to gene over-expression that modulate components of the slit diaphragm. An injury model would align more with transient chemical or hydrostatic pressure provocations. Perhaps avoid using the term injury.
15. Mechanical force is mentioned in the discussion yet there is no methodological approach that addressed forces being exerted on nephrocytes nor the effect this had on nephrocyte function or phenotype. Without such a provocation, it is hard to state that the modifications to nephrocytes were linked to mechanotransduction.
16. Figures 4 and 5 have no images of nephrocytes to support the quantified data. Can these be presented?

Reviewer #1 (Comments to the Authors (Required)):

Major comments

Cheerio localization:

In Figure 1 and associated text, the authors conclude that Cheerio partially colocalizes with the slit diaphragm protein Duf, which they suggest may indicate Cheerio is in complex with slit diaphragm-associated proteins. However, the images provided (Figure 1A,B) are very low magnification, which makes determining its precise localization difficult. Can the localization of Cheerio be determined in high-resolution images similar to those in panel E? This would be particularly useful in examining the localization of the wildtype, inactive, and active forms of the protein. Similar high magnification images would be helpful for Human FilB, which they suggest also localizes to slit diaphragms (Supplemental Figure 4A). If these types of images cannot be provided, it seems necessary to significantly tone down the conclusion that these proteins are located at slit diaphragms.

We added higher resolution microscopy images for the endogenously tagged Cheerio, the different Cheerio variants and the human Filamin B expressing nephrocytes.

High resolution imaging using confocal microscopy combined with Airyscan revealed co-localisation of Duf with endogenously tagged Cheerio (Cher:GFP) in garland cells isolated from 3rd instar larvae. The newly generated images are now included in **Figure 1B** of the manuscript.

Supp. Figure 1: Cheerio variants. **A** Immunofluorescence staining confirmed the expression of Cheerio (green: Cheerio:GFP combined with anti-GFP antibody; magenta: anti-Cheerio antibody) in *Drosophila* wildtype embryos, which are labelled with Duf (cyan). Scale bar = 50 μ m.

Our data reveals different localisation patterns for the three variants. While Wildtype and inactive Cheerio can be found largely in the cytoplasm, active Cheerio showed an

alternating pattern with Duf confirming our hypothesis that Cheerio in particular the active version localizes to the slit diaphragm complex. The data is integrated in **Supp.**

Figure 1.

Supp. Figure 1: Cheerio variants. C High-resolution microscopy using Airyscan together with confocal microscopy revealed localisation of active Cheerio to the nephrocyte diaphragm (represented by Duf (cyan)), while wildtype and inactive Cheerio also present with a more cytoplasmic localisation. Scale bar = 5 μ m. Control: *w;sns-Gal4;UAS-dicer2*, Cher wildtype: *w;sns-Gal4/UAS-cher-wildtype-MSR;UAS-dicer2/+*, Cher Inactive: *w;sns-Gal4/+;UAS-dicer2/UAS-cher-inactive-MSR*, Cher Active: *w;sns-Gal4/+;UAS-dicer2/UAS-cher-active-MSR*.

We also performed high resolution imaging with Airyscan and confocal microscopy for nephrocytes expressing the human Filamin B wildtype construct. The human Filamin B has a C-terminal HA tag, which was used for co-localisation studies with Duf. We did not observe a co-localisation of human Filamin B with Duf, which is in contrast to Cheerio (see above). However, the HA pattern resembles the nephrocyte diaphragm pattern we observe for Duf and Pyd, hence human Filamin B does localize to the nephrocyte diaphragm. Whether there is a direct interaction with another nephrocyte diaphragm protein remains unknown. We have amended the manuscript to reflect these new data.

Supp. Figure 4: Human Filamin B localizes to the nephrocyte diaphragm and results in only a mild nephrocyte phenotype. D High resolution imaging revealed human Filamin B wildtype localizes close to the nephrocyte diaphragm, which is visualized by anti-Duf staining (cyan). human Filamin B wildtype has a C-terminal HA tag. HA: magenta Scale bar = 5 μ m.

The methods section (Table 2) suggests the three Cheerio MSR variants from the Huelsmann 2016 paper are UAS, however the description in that paper indicates they are knock ins at the endogenous locus (attB-mediated replacement of the 3' end). If this is correct, it will change the wording in places. For example, the text indicates these constructs are being driven by Sns-Gal4, but instead they would be expressed from the endogenous cheerio promoter. It may also be more accurate to indicate that expression of the Cheerio-MSR Active form of the protein causes hypertrophy, as opposed to "elevated" levels of Cheerio-MSR Active protein.

We do apologize for the confusion. The fly strains used in this manuscript are indeed UAS-driven and were kindly provided by Jari Yläanne. They are not mentioned in the Huelsmann et al, 2016 manuscript, but the mutations described in this manuscript are the same. We changed this in the manuscript.

There are notable differences in the localization patterns of Cheerio:GFP and the anti-Cheerio staining, particularly in panel A. Is there an explanation for this? They appear more similar (cortical) in panel B.

We agree with the reviewer, that there are notable differences in localisation of Cheerio, in particular in embryonic nephrocytes. A few explanations could be an unspecific binding of the Cheerio antibody, which causes an additional staining pattern, when compared to Cheerio:GFP. We used an anti-GFP antibody to enhance the

Cheerio:GFP signal and the antibody efficiency of the anti-Cheerio and the anti-GFP antibody might be different. The localisation pattern is much more similar in larval garland cells, which are used for all experiments in the manuscript.

I am also confused by the wildtype Cheerio:GFP localization patterns in panel B and C. I believe the Cheerio:GFP in B is the GFP trap line mentioned in the methods section (this was not clear in the figure legend). Is there an explanation for why the wildtype Cheerio:GFP in panel B appears cortical while the Cheerio:GFP with wildtype MSR (essentially the wildtype protein I believe) is cytoplasmic?

We apologize for the confusion and changed the figure legend accordingly. One potential explanation for the different localisation pattern might be the overexpression of Cheerio in panel C. We used UAS-driven Cheerio variants, which resulted in higher expression levels, than the endogenous Cheerio. For the GFP-trap line we expect endogenous levels. Hence, the increased protein levels seem to result in an additional cytoplasmic localisation. Moreover, the UAS-driven Cheerio WT is lacking the actin-binding domain, which might also partially impact on the localisation of Cheerio. This potential impact seems to be of minor importance, as the active variant, which is also lacking the actin-binding domain appears to be cortical. We added these observations into the results section.

*Cheerio exerts a protective effect in damaged nephrocytes:
Figure 2. Panel D suggests that Cheerio-Active can rescue FITC:Albumin uptake caused by knockdown of slit diaphragm proteins. Knockdown of these proteins disrupts the pattern of slit diaphragms on the nephrocyte surface, and presumably reduces uptake of fluorescent tracers due to subsequent loss of labyrinthine channel area, where the majority of endocytic uptake of these tracers occurs. However, the current data (Figure 2B) suggest that slit diaphragms are not restored by Cheerio-Active. Is Cheerio-Active directly affecting rates of endocytosis through some other mechanism? It would be helpful if the authors could provide more discussion of possible mechanisms underlying these findings.*

To address, whether Cheerio, in particular the active variant, influences endocytosis, thereby causing a functional rescue of the Duf and Sns knockdown phenotype, we performed additional experiments and literature research.

We agree, that the FITC-Albumin uptake assay cannot be used to distinguish between filtration at the nephrocyte diaphragm and uptake/endocytosis in the lacunae. Hence, we do not know the exact mechanism, why and how the rescue fly strains (Cheerio Active and Duf/Sns RNAi) perform better than single knockdown flies. We tried to address this question in more detail. Our data describing the effects of overexpression of the Cheerio variants (**Figure 1G**) does not show an increase of FITC-Albumin

uptake in either of the genotypes, suggesting no direct effect on endocytosis. Moreover, changes of the slit diaphragm and the lacunae structure might be very subtle and might require higher resolution microscopy such as electron microscopy. Cheerio exhibits an actin-binding domain and is an actin cross-linker. Therefore, the overexpression of active Cheerio might stabilize the foot processes and lacunae and result in a better result in the FITC-Albumin uptake assay.

To investigate the link between endocytosis and Cheerio we also performed Rab7 stainings in Cheerio depleted nephrocytes, as those present with an increased filtration function (Koehler *et al*, 2020). Rab7 is described to be involved in the endocytic pathway in the transition from late endosome to lysosomes (Vanlandingham & Ceresa, 2009) and was shown to be one of the important Rab GTPases in nephrocytes (Fu *et al*, 2017). In nephrocytes, Rab7 seems to mediate vesicle trafficking pathways resulting in protein degradation and membrane recycling. Loss of Rab7 caused a decreased uptake of ANF-RFP and AgNO₃ in adult nephrocytes and severe morphological changes in larval nephrocytes such as absence of lysosomes, reduced lacuna size and deformed nephrocyte diaphragms (Fu *et al*, 2017). We used a commercially available antibody against Rab7 and assessed the effects of Cheerio loss on Rab7 positive vesicles. In detail, we measured the Rab7 intensity representing Rab7-positive vesicles in nephrocytes and could show that loss of Cheerio did not result in changed levels of Rab7-positive vesicles.

Supp Figure 3

Supp. Figure 3: FITC-Albumin uptake assays and Rab7 immunofluorescence staining of Cheerio rescue strains. **C** Overexpression of either inactive or active Cheerio in nephrocytes did not cause any changes in Rab7 mean intensity. Duf: cyan; Rab7: magenta; GFP: green. Scale bar = 25 μ m. **D** Immunofluorescence stainings in control and Cheerio kd nephrocytes did not reveal changes in Rab7 mean intensity. HRP: green; Rab7: magenta. Scale bar = 25 μ m.

We also performed Rab7 immunofluorescence stainings in nephrocytes overexpressing either inactive or active Cheerio. Neither of the variants caused a significant increase of Rab7-positive vesicles. This data is now included in **Supp. Figure 3C,D**. Based on these findings we concluded, that the filtration rescue caused by active Cheerio expression is not induced via Rab7-mediated endocytosis.

To further investigate the link between Cheerio and endocytosis, we also searched literature and found a few links of Cheerio and Filamin A with endocytosis. Filamin A interacts with Big2 regulate endocytosis (Sheen, 2014), the same was observed for the physical interaction of Filamin A with formin 2 (Lian *et al*, 2016). In addition, Filamin A associates with caveolin-1, which promotes vesicle internalization and trafficking (Sverdlov *et al*, 2009). A previous study in *Drosophila* neurons revealed an interaction of Cher and varicose resulting in pebble mediated endocytosis via Rho1 activation and actin cytoskeleton reorganisation (Jeong *et al*, 2017). While there are a few links of Filamin A/ Cheerio and endocytosis our data investigating Rab7 positive vesicles did not reveal a significant change in endocytosis upon depletion or overexpression of Cheerio. Hence, we concluded that our FITC-Albumin results primarily reflect filtration function rather than endocytosis.

In their previous study (Koehler et al., JASN 2020), Cheerio knockdown had no apparent effect on otherwise normal nephrocytes. The model proposed here suggests that Cheerio has a protective effect in injured podocytes/nephrocytes. Thus, if Cheerio protects nephrocytes in response to injury, one would predict that knockdown of Cheerio in a damaged nephrocyte should significantly dampen that protective response, leading to stronger defects, be it cell morphology, size, or function. If the authors agree with this rationale, can they perform a double knockdown of Cheerio and Sns or Duf to assess whether this is correct?

We agree with this rationale and thank the reviewer for this comment. We generated double knockdown flies (*w;Sns-Gal4/+;UAS-Cher-RNAi/UAS-Duf-RNAi* and *w;Sns-Gal4/+;UAS-Cher-RNAi/UAS-Sns-RNAi*) and performed morphological and functional assessment of nephrocytes. As previously published by us, Cheerio knockdown did not result in a morphological phenotype (**Figure A, below**), while nephrocyte size was significantly decreased and FITC-Albumin uptake was increased (**Figure B,C**)

(Koehler *et al*, 2020). Loss of Duf and Sns caused a severe morphological phenotype (**Figure A**), a significant size decrease (**Figure B**) and a significantly disrupted filtration function (**Figure C**). Combining Cheerio RNAi with either Duf or Sns RNAi together did not result in an additional phenotypic effect in regard to filtration or morphology. However, nephrocyte size was further decreased suggesting a worsening of the phenotype by simultaneous depletion of Cheerio with the ND proteins. As morphology is already affected very strongly in the Duf and Sns knockdown an additional effect induced by Cheerio loss might not be detectable. Moreover, loss of Cheerio resulted in an increased FITC-Albumin intensity, while depletion of the ND proteins caused a significant reduction. The combination of both RNAis caused a decreased FITC-Albumin intensity when compared to control cells, but was still (significantly) increased in comparison to the Duf and Sns single knockdown (**Figure C**). Hence, it is very difficult to interpret the data and whether the simultaneous depletion results in a worsening of the filtration defect. Although the data shown here investigating nephrocyte size suggests a worsening of the phenotype after depletion of Cheerio, we decided to not include the data in the manuscript, as the results obtained from the FITC-Albumin assay are difficult to interpret. However, we wanted to show our results and explain our decision in the reply to the reviewers.

Figure: Simultaneous depletion of Cheerio and Duf/Sns. **A** Immunofluorescence staining using a Duf (cyan) and Pyd (magenta) antibody reveals no phenotype in Cheerio depleted cells (Cher kd), while Duf and Sns single knockdown as well as Cheerio and Duf/Sns double knockdown presents with a severely disrupted morphology. Scale bar = 25 μm and 5 μm . **B** Nephrocyte size is decreased in Cheerio, Duf and Sns single knockdown nephrocytes. Combining Cheerio RNAi with either Duf or Sns RNAi results in an additional and significant size decrease. One-way ANOVA plus Tukey's multiple comparisons test: ***: $p < 0.001$ ****: $p < 0.0001$. **C** Filtration function was assessed by FITC-Albumin assays and reveals an increased FITC-Albumin intensity in Cheerio depleted nephrocytes (Cher kd), while loss of Duf and Sns results in a significant decrease of the FITC-Albumin intensity. The combination of either Cheerio and Duf RNAi or Cheerio and Sns RNAi causes a decreased FITC-Albumin intensity when compared to control cells, but also a significantly higher intensity when compared to the Duf and Sns single knockdown cells. One-way ANOVA plus Tukey's multiple comparisons test: ***: $p < 0.001$.

TOR and Yki are downstream of Cheerio: Page 21, line 16. The subsection title of "TOR and Yorkie signaling induce nephrocyte hypertrophy and appear to be downstream targets of Cheerio" seems an overly strong conclusion because, as the authors note in the manuscript, the expression of TOR DN or HIPPO alone decrease cell size, thus the apparent rescue of Cher Active hypertrophy may simply be the averaged effects of two independent processes acting on cell size. It may support their conclusion if they could show increased activity for these pathways in Cher Active cells. There are available genetic reporters for Hippo activity (Parker and Struhl, PLOS Biology 2015). TOR activity has been assessed in flies using various readouts, such

as phospho-4E-BP antibody staining (Teleman et al., Developmental Cell. 2005). Similarly, based on the current data, the proposed relationship between Cheerio and TOR/Yki in Figure 6 should be made less conclusive; perhaps by making the arrows dashed or adding question marks. The figure legend text should also reflect that these relationships are somewhat tentative.

We agree that it is of greatest interest to investigate whether expression of active Cheerio indeed activates either TOR or HIPPO or both pathways. Hence, we performed additional experiments in flies expressing the different Cheerio variants. To assess TOR activation, we did immunofluorescence stainings in nephrocytes using a phospho-4E-BP1 antibody and quantified intensity in the different genotypes. Our data (**Figure 4H**) revealed no significant differences of phosphorylated 4E-BP1 levels upon expression of inactive or active Cheerio when compared to controls. Performing Yorkie immunofluorescence staining revealed an increase of nuclear localisation of Yorkie upon expression of active Cheerio, while there was no significant change in nephrocytes expressing inactive Cheerio (**Figure 4I**). This data shows an activation of Yorkie in active Cheerio expressing cells, while TOR signalling does not seem to be activated. To further investigate, whether the pathways act downstream of Cheerio, we also performed additional genetic interaction experiments. In detail, we combined active Cheerio with overactive Yorkie or overactive TOR. By doing so, we addressed whether a combined overactivation results in a further size increase of nephrocytes. We hypothesised that if Yorkie and/or TOR are in the same pathway as Cheerio no additional size increase should be observed in nephrocytes. Indeed, co-expression of active Cheerio and overactive Yorkie did not cause an additional size increase (**Figure 4J**). However, expressing active Cheerio and overactive TOR resulted in a significant size increase when compared to active Cheerio expressing cells (**Figure 4J**). Taken together, our data therefore shows, that TOR does not seem to be activated by Cheerio and does not seem to be a downstream target. Yorkie translocates into the nucleus upon active Cheerio expression and seems to be a downstream target of Cheerio. We added the new data to the manuscript and also changed the graphical summary in **Figure 6** accordingly. We added the new data and conclusion in the manuscript. These experiments were very informative and have helped us hone in on the Cheerio pathway; we thank the reviewer for these suggestions.

Figure 4

Figure 4: TOR and Hippo signalling mediate the hypertrophy phenotype in nephrocytes. **H** Immunofluorescence stainings using an antibody against phosphorylated 4E-BP1 did not reveal any significant differences in mean intensity in nephrocytes expressing active or inactive Cheerio in comparison to control cells. **I** Immunofluorescence staining using an antibody against Yorkie revealed a significantly higher mean nuclear intensity in nephrocytes expressing active Cheerio when compared to control cells. One-way ANOVA plus Tukey's multiple comparisons test: ****; $p < 0.0001$. **J** Nephrocyte size was significantly increased when combining active Cheerio with over-active TOR, while the combination of active Cheerio with over-active Yorkie did not cause a significant size increase. One-way ANOVA plus Tukey's multiple comparisons test: **; $p < 0.01$.

Figure 6: The mechano-protective role of Cheerio in *Drosophila* nephrocytes. Cheerio is active in diseased nephrocytes and accumulates at the cell periphery. This activation and the increased protein levels result in a mechano-protective effect via hypertrophic growth, mediated via TOR and Hippo signalling. However, Cheerio levels need to be tightly controlled, as excessive increase of protein levels results in a shift from the protective to a pathogenic effect, including a morphological and functional phenotype.

Minor comments

Figure 1 is titled: "Elevated Cheerio-MSR levels result in nephrocyte hypertrophy", however, because the hypertrophy only results from the active form, perhaps more specific language could be used, such as "Elevated levels of Cheerio-MSR-Active result in nephrocyte hypertrophy".

We changed the title of Figure 1 accordingly.

Figure 2 The order of genotypes in the images in panel B is different from the x-axis of graphs in panels C/D. Similarly, the order of genotypes within panels C/D is different from the x-axis in F. Since these include all of the same genotypes, it would be helpful to the reader if the order of genotypes could be kept consistent. The order of genotypes in these graphs is also different from that in Supplemental Figure 2. This makes it confusing when trying to compare the effects of active vs inactive Cheerio in the different genotypes.

It is not possible to distinguish the genotypes by symbols in Figure 2 panel D. Can the symbols and lines also be color coded or the symbols enlarged or otherwise made more obvious?

The figure legend for panels E,F state that Cheerio Active rescues delayed pupation of Duf or Sns RNAi, but the data only indicate a delayed pupation for Duf (and associated rescue). Thus it seems reference to a rescue of Sns RNAi should be omitted here.

We agree and changed the order of genotypes and symbols as proposed.

We also changed the figure legend for panels E and F, as there is no significant rescue of the Sns phenotype.

Figure 3 In Panel B, it appears that expressing hFilB in Duf kd cells is somehow restoring Duf expression. Is this correct? If so, can the authors provide any explanation?

In the images shown it might look like there is more Duf in the FilB rescue nephrocytes. However, we think this is mainly due to the strong background signal we also see in the surrounding tissue. The same unspecific background staining can be observed in the wildtype control and the Sns rescue. The comparison between the wildtype and the Duf rescue shows a clear loss of Duf expression.

Of note, the human Filamin B construct contains the actin-binding domain, which is lacking in the Cheerio variants used in Figure 2 and Supp Figure 2 for the rescue experiments. It is conceivable, that the presence of the actin-binding domain might impact on Duf expression and localisation.

*In Panel D, there is a "****" but no line indicating which genotypes are statistically different. The figure legend suggests none of them are different, so the "****" probably just needs to be removed.*

We removed the '****'.

Figure 4 Can the data in panels F and G be merged together with D and E, respectively, to make comparisons between all genotypes more straightforward?

We could merge the files, but decided to not do this for the following reasons. Experimental setups are different. Due to the GFP-tag of the Cheerio variants, the rescue with either TOR-DN or HIPPO were performed using the FITC-Albumin in combination with the anti-Albumin antibody, while TOR-DN and HIPPO alone were done with the normal FITC-Albumin assay protocol. This resulted in additional incubation times and procedures for the rescue cells; hence the absolute values cannot be compared directly, but they could be compared after normalisation to the respective control. We merged the normalized data-sets and show them below. Performing statistics here is critical, as these are independent experiments with different protocols as mentioned above. If the reviewer and the editors think, merged data-sets would be better, we can of course provide them for the normalized data and change the figure accordingly.

Figure: Active Cheerio mediated hypertrophy is reversed by blocking TOR and activating HIPPO signalling. **A** Cher Active causes a significant size increase when compared to inactive Cheerio, which can be reversed by blocking TOR signalling (TOR DN). However, blocking TOR alone also results in a significant size decrease. One-way ANOVA plus Tukey's multiple comparisons test: ***, $p < 0.001$, ****, $p < 0.0001$. **B** Activating Hippo resulted in a rescue of the hypertrophy phenotype observed in nephrocytes expressing active Cheerio. Activating Hippo alone also caused a significant size decrease. One-way ANOVA plus Tukey's multiple comparisons test: ****, $p < 0.0001$.

Page 19, line 8 indicates that the Cheerio constructs do not possess the ACB, but as discussed above, I do not believe this is accurate. The mutant Cheerio knock-ins described in the Huelsmann 2016 paper suggest the MSR variants should be part of the full-length protein (long isoform), which should therefore contain the ACB. I believe this also affects the interpretation of the data as presented on page 29, line 10, where the inference is made that hFilB rescues cells size due to possessing the ACB, thus the ACB is important for cell size, however if it is correct that the Cheerio variants are indeed full-length proteins possessing the ACB, this conclusion does not seem supported.

Again, we do apologize for make this mistake. The fly strains used are UAS-driven strains without the actin-binding domain. They are not described in the mentioned manuscript (Huelsmann et al., 2016). We changed the manuscript accordingly.

Yki is the common abbreviation for Yorkie, not "Yrk".

We changed that accordingly.

Page 24, the text does not make it clear that the phenotypes result from homozygosity for the Cheerio MSR Active or Inactive variants, not wildtype Cheerio.

We rephrased the text to make our approach clearer and to emphasize that we only generated homozygous flies for the active and inactive variant. The wildtype construct was not assessed.

Table 2 lists UAS-hFilB constructs delta ACB and delta MSR, but I do not believe there are any data or discussion of these lines in the manuscript. Probably remove these from table, along with associated primer information.

We are sorry for this mistake and deleted the information.

Reviewer #2 (Comments to the Authors (Required)):

The paper describes the effects of modulating the fly ortholog of human filamins. This is done in fly nephrocytes, which are used to study aspects of human kidney podocytes and proximal tubule cells. The paper focuses on nephrocytes as a model of podocytes but uses endocytic function as a marker, a model of proximal tubule cell function. Nephrocyte are both a model of both human podocyte and proximal tubules cell and this should really be mentioned. Also, the localisation of filamin B in human proximal tubules cells might suggest the drosophila model is better suited to establishing filamin's role in tubules cell endocytosis - given that its at the cell surface. Its not so simple to say that Drosophila filamin (Cher) models only filamin biology in human podocytes. Apologies in advance if I've misunderstood or missed something I should have seen or considered.

We agree, that there are evidences for similarities between nephrocytes and proximal tubular cells. We included these findings and our thoughts in the discussion.

1. The paper deals with Filamin's 'mechano-protective role' in nephrocytes however there needs to be experimental data presented in the paper to support this mechanotransduction role. At present there is no experimental data relating mechanotransduction, the mechano-protective role is inferred from other studies and not demonstrated directly.

We agree with the reviewer and performed additional experiments. We induced mechanical stress via hypo-osmotic stress and could show that nephrocytes respond with swelling and a significant size increase in control cells (**Figure A below**).

Figure: Hypo-osmotic stress causes nephrocyte swelling, but does not impact on morphology. **A** Nephrocyte size increased significantly after 5mins hypo-osmotic stress. Student's t-test: ***: $p < 0.001$. **B** Confocal microscopy visualizing the nephrocyte diaphragm protein Duf did not reveal obvious morphological changes after 5mins hypo-osmotic stress. Scale bar = 25 μm .

We hypothesized to see morphological changes as well, which might be rescued in nephrocytes expressing active Cheerio. However, we did not observe changes in morphology upon hypo-osmotic stress in control cells (**Figure B above**), which is most likely due to the short time frame of the experimental setup; 5mins stress and immediate fixation afterwards. Also, changes might be very subtle and cannot be visualized by normal confocal microscopy as done here.

However, we decided to keep the term 'mechano-protective' as the Cheerio variants we used in our experiments express the mechanosensor domain and lack the actin-binding domain. Moreover, the active mechanosensor variant resulted in the best rescue and the putative protective hypertrophy phenotype. Based on our findings we argue that these protective effects are elicited by the activation of the mechanosensor region. Hence, we kept the term mechano-protective.

2. Neither the introduction nor the discussion mentions a considerably important paper about YAP/TAZ signaling in mammalian podocytes and Drosophila nephrocytes by Hurcombe et al Nature Communications 2019. That paper's findings need to be mentioned and the current data examined within the context of Hurcombe's work.

We agree that this is a very interesting paper and examined our data in the context of the findings of Hurcombe et al.. In their paper they investigated the role of GSK3 β /shaggy in podocytes and nephrocytes and could show that loss of shaggy causes absence of nephrocytes, similar to what we observe when blocking WNT (wingless) signalling. Under normal conditions shaggy phosphorylates β -catenin/armadillo, which then becomes ubiquitinated and degraded, resulting in abrogation of Wg signalling. Loss of shaggy therefore causes accumulation of armadillo in the cytoplasm and subsequent translocation into the nucleus, resulting in activation of Wg targets. Hurcombe et al's data show that over-activation of Wg is detrimental for nephrocyte development and results in their absence. We observed the same phenotype for the opposite manipulation: blocking Wg signalling by expressing a constitutively active repressor version of pangolin (LEF/TCF), which prevents the activation of Wg targets causes loss of nephrocytes. Taken together, our data and the data from Hurcombe confirm findings from mammalian models, showing that WNT signalling is critical for podocyte/nephrocyte development and needs to be tightly controlled (**Figure A,B,C below**).

Figure: Wingless and Hippo signalling in nephrocytes. **A** Depletion of shaggy (GSK3 β) caused the absence of nephrocytes (Hurcombe *et al*, 2019). **B,C** Repressing Wingless (WNT) signalling resulted in the absence of nephrocytes as observed in confocal microscopy (**B**) and FITC-Albumin (**C**) assays.

In their manuscript, Hurcombe et al. provide data showing that HIPPO signalling is responsible for the pathology observed upon loss of shaggy.

In detail, proteomic analysis revealed an upregulation of Ajuba and YAP/TAZ targets after depleting GSK3 β . Ajuba blocks phosphorylation of YAP/TAZ, which causes translocation of the two proteins into the nucleus, activation of downstream signalling and switching HIPPO off. Verteporfin was used to prevent translocation of YAP/TAZ into the nucleus in shaggy depleted and LiCL treated nephrocytes and had a beneficial effect on the cells. Taken together, the phenotype observed upon loss of shaggy is YAP/TAZ mediated, as switching HIPPO on is beneficial in this context.

Our data show that hyperactive Yorkie (YAP), which switches HIPPO off, results in a hypertrophy phenotype (**Figure 4B**), while blocking Yorkie by expressing HIPPO

caused a nephrocyte size decrease (**Figure 4G**). Moreover, the activation of HIPPO reversed the increased cell size in nephrocytes expressing active Cheerio (**Figure 4D**). In addition, we provide data showing that Yorkie translocates into the nucleus upon activation of Cheerio, suggesting Yorkie is a downstream target of Cheerio.

Taken together, both studies show that HIPPO signalling plays an important role in nephrocyte biology and is downstream of GSK3 β /shaggy and Cheerio. We summarized these findings and conclusion in the discussion.

3. Nephrocyte numbers should be stated for all experimental genotypes. If modulations are protective then nephrocyte numbers should be similar - this needs to be shown.

For our experiments and analysis, we did a cell-by-cell analysis. This means, that we measured nephrocyte size and fluorescent intensity for every single cell in this image. The individual values are then used to calculate the mean for each fly, which means that one data point is representing one fly. We added this in the material and methods section. Based on our analysis approach, we think numbers do not need to be equal to be able to observe protective effects as we measure and quantify single cells and compare mean values between flies and genotypes.

However, we assessed the mean number of nephrocytes per fly after expressing the different Cheerio variants and did not observe any significant differences in nephrocyte number (**Figure A below**). We also assessed whether loss of Duf and Sns resulted in less nephrocytes and whether this can be rescued by expression of active Cheerio (**Figure B below**). We did not observe any significant changes in any of the genotypes assessed.

Figure: Nephrocyte numbers. **A** Nephrocyte number was assessed in control nephrocytes and cells expressing the three different Cheerio variants. Over-expression of the Cheerio variants did not result in a significant change of nephrocyte number per fly. **B** Nephrocyte number was not changed upon loss of Duf and Sns as well as in the active Cheerio rescue flies.

We apologize if we misunderstood the comment from the reviewer.

4. Nephrocyte area need to be stated in square microns, not as relative values.

We added all graphs showing raw data (microns) in **Supp. Figure 7**. We decided to keep the normalized relative values for easier comparison. However, we agree that it is of interest to show the absolute values. As experimental setups were different due to the GFP-tag of the Cheerio variants, it is difficult to compare raw data throughout. In experiments which included the Cheerio variants, we performed the FITC-Albumin uptake assay and subsequent anti-Albumin immunofluorescence. This caused additional incubation times and solutions and has an impact on the FITC-Albumin signal.

Supp. Figure 7: Absolute values of nephrocyte size. A-N For two variables we used Student's t-test and considered a p-value < 0.05 as significant. For more than two variables we used One-way ANOVA plus Tukey's multiple comparisons test and also considered a p-value < 0.05 as significant.

5. Figure 1 and other images. The images of nephrocytes are very small, most of the image is extraneous area and the images do little to support the quantified data.

From the images, the nephrocytes do not appear to be larger in the Cher active experimental group.

We agree, that in some of the pictures we have a lot of extraneous area, but we decided to keep the magnification equal for all genotypes for comparison. Often one of the images cannot be cropped. We showed a different representative image for Cher Active. We also added magnifications or high-resolution imaging where it is possible to support our quantification data.

6. There is a large discrepancy between the albumin binding as quantified in figure 3 compared to albumin binding for similar genotypes in Figure 2 (70-80% reduced in figure 3 vs 30% reduced in Figure 2 for both the sns KD and duf KD flies).

This discrepancy is a result of different experimental setups. The Cheerio variants are fused to a GFP tag, hence the FITC-Albumin cannot be quantified after incubation, as the GFP signal is overlapping with the FITC signal. Therefore, we performed Albumin immunofluorescence stainings (with anti-Albumin) after incubation with FITC-Albumin. The anti-Albumin antibody was then labelled with a Cy3 secondary antibody to enable quantification of Albumin within the cell. The Filamin constructs are fused to a HA-tag, therefore FITC-Albumin assays could be performed without additional antibody-based staining. The genotypes (Duf kd and Sns kd and respective controls) were done in the same way (Figure 2 plus additional antibody-based staining, Figure 3 without additional staining) to enable comparisons with the respective rescues. For normal FITC-Albumin uptake assays isolated nephrocytes were incubated in FITC-Albumin for 1 min, followed by a 1 min washing step and subsequent fixation for 20 mins and mounting. To perform antibody based stainings, nephrocytes were also Incubated with FITC-Albumin for 1 min, followed by a washing step for 1 min and fixed for 20 mins with formaldehyde. Instead of mounting cells were then incubated for 1 h with methanol, followed by 3 washing steps for 10 mins and incubation with the albumin antibody overnight. The next day, cells were again washed for three times, incubated with the Cy3 secondary antibody for 45 mins, and washed again before they were mounted. Due to this long and intense staining protocol, the Albumin signal will be lower, if compared to cells immediately fixed after incubation with FITC-Albumin. Moreover, due to weak antibody binding of the Albumin antibody, signal intensity was much lower. We think that these different protocols are causative for the discrepancy observed. As all

genotypes, which we compare, were done with the same protocol, this should not be a problem.

7. Figure 3B the images should be presented in the same order as the data in the graphs (Fig 3C and D).

We changed that accordingly.

8. Figure 3D the asterisks needs an accompanying line.

We deleted the asterisks, as there was no difference between the rescue and single knockdown cells. The significance described the effect resulting from comparing the control (h Fil B WT) with the rescues and single knockdowns. For simplification, we removed the asterisks.

9. Figure 3. Cells really need to be counterstained with some general cell marker. They cannot be seen in the Duf KD and other images.

To make the cells more visible we added circles to mark the outlines of the cells.

10. Figure 3's figure legend: this is a little confusing me: the title states 'Human Filamin B rescues nephrocyte size and function', then for 3B it states 'Expression of human Filamin B wildtype (human Fil B WT) in Duf and Sns depleted nephrocytes did not restore morphology', then 3C states, 'Nephrocyte size was restored by expressing human Filamin B wildtype, then 3D states 'FITC Albumin uptake is not rescued by expression of the human Filamin B wildtype. If I've understood this, it looks like Human Filamin B rescues the AgNO3 toxicity independently of rescuing albumin binding / endocytosis; which is odd. Yet function is stated as being restored - which is only partly the case.

We changed the figure title and legend to make our observations and the resulting conclusion clearer.

11. Supp Fig 4 -colocalization data is required to support the claim that Human filamin B localises to the slit diaphragm. Currently it shows the HA tag and Pyd are at the cell surface but that's far from showing colocalization at the slit diaphragm. If the same logic was applied more generally, all cell surface proteins could be claimed to localise to the slit diaphragm, and that is not the case.

In the first description of the nephrocyte diaphragm by Weavers et al. Pyd was identified as a component of the nephrocyte diaphragm (Weavers et al, 2009). We performed additional stainings combining confocal microscopy with Airyscan to get high resolution images of human Filamin B and the nephrocyte diaphragm protein Duf. The data is included in **Supp. Figure 4D** and below. As the human Filamin B construct

contains a C-terminal HA tag, we used the tag for visualisation. In contrast to Cheerio, human Filamin B does not co-localize with Duf. However, human Filamin B might interact with other components of the nephrocyte diaphragm such as Sns or Mec2 (dPodocin). We could not investigate this in greater detail as we do not have working antibodies against Sns and Mec2. However, the expression pattern of human Filamin B looks similar to what we observe for Duf and Pyd. Hence, we concluded that human Filamin B localizes to the nephrocyte diaphragm. However, we toned this down to 'localizes close to the ND'.

Supp. Figure 4: Human Filamin B localizes close to the nephrocyte diaphragm and results in only a mild nephrocyte phenotype. D High resolution imaging revealed human Filamin B wildtype localizes close to the nephrocyte diaphragm, which is visualized by anti-Duf staining (cyan). human Filamin B wildtype has a C-terminal HA tag. HA: magenta Scale bar = 5 μ m.

12. Figure 5. The control appears to comprise a line with two chromosome 3 balancers (MKRS/Tm6B). That's a lot of mutations that cannot be regarded as representative of wild type and therefore another control really needs to be used.

We agree, that the ideal control is a homozygous Sns-Gal4 line without the Dicer and without any balancers. However, we failed to generate this fly strain.

We compared the absolute values for the nephrocyte size of $w;sns-Gal4/+;UAS-dicer2/+$ and $w;sns-Gal4; MKRS/TM6B$ (control) and did not see a significant difference. We added this genotype to the graph in Figure 5A and 5D. We also did not

observe a difference in size when we compared the homozygous Cher Inactive and homozygous Cher Active flies, supporting or finding that the hypertrophy phenotype is lost with increasing Cher Active expression. In addition, the presence of the balancers also did not impact on morphology as shown in Supp Figure 5A.

Also, considering that the flies without the balancers would likely be healthier, the effect observed in the FITC-Albumin assay would be even bigger, which means homozygous Cheerio flies would present with an even worse phenotype. As a second functional read-out we performed the AgNO₃ toxin assay. Here we compared the homozygous flies to their respective heterozygous controls. We did the same for the homozygous human Filamin B WT flies and experiments and found no significant difference in cell size when comparing *w;sns-Gal4/+;UAS-dicer2/+* and *w;sns-Gal4; MKRS/TM6B* (control). The above-mentioned argument regarding the FITC-Albumin assay also applies for the Filamin B expressing flies.

Figure 5

Figure 5: Excessive increase of Cheerio and Filamin B levels results in a pathological response.

A Homozygous Active or Inactive Cheerio were used to increase the level of protein expression in nephrocytes. Comparison with control flies (*w;sns-Gal4/+;UAS-dicer2/+* and control: *w;sns-Gal4;MKRS/TM6B*) revealed no difference in nephrocyte size. Cher Inactive: *w;sns-Gal4;UAS-cher-inactive*; Cher Active: *w;sns-Gal4;UAS-cher-active*. **B** FITC-Albumin uptake was significantly decreased in both genotypes, inactive and active Cheerio. One-way ANOVA plus Tukey's multiple comparisons test: **: $p < 0.01$; ***: $p < 0.001$. **C** The AgNO₃ toxin assay revealed a severely delayed pupation in homozygous flies when compared to their heterozygous controls. Cher Inactive het: *w;sns-Gal4/+;UAS-cher-inactive/UAS-dicer2*; Cher Active: *w;sns-Gal4/+;UAS-cher-active/UAS-dicer2*; Cher Inactive homo: *w;sns-Gal4;UAS-cher-inactive*; Cher Active homo: *w;sns-Gal4;UAS-cher-active*. Two-way ANOVA plus Tukey's multiple comparisons test: **: $p < 0.01$; ***: $p < 0.001$; ****: $p < 0.0001$. **D** Homozygous

expression of human Filamin B wildtype resulted in a significant size decrease. Student's t-test: *: $p < 0.05$. *w;sns-Gal4/+;UAS-dicer2/+* Control: *Sns-Gal4;MKRS/TM6B*; *w;sns-Gal4;MKRS/TM6B*; hFil B WT: *w;sns-Gal4;UAS-hFilamin B WT*. **E** FITC Albumin uptake was severely impaired by homozygous expression of human Filamin B WT when compared to controls. One-way ANOVA plus Tukey's multiple comparisons test: ***: $p < 0.001$. **F** Comparison of homozygous and heterozygous controls in the AgNO_3 toxin assay also revealed severe filtration defects. Two-way ANOVA plus Tukey's multiple comparisons test: *: $p < 0.05$; ***: $p < 0.001$.

In addition, to further confirm our findings we performed experiments at 28°C to increase Cheerio levels. Here, we used *w;sns-Gal4/+;UAS-dicer2/+* flies as control, which is the correct one. We observed a severe morphological phenotype in flies expressing the different Cheerio variants (Supp. Figure 5B).

Therefore, we decided to keep the data in the manuscript.

13. Discussion: Filamin B is expressed at the apical surface of human proximal tubule cells - cells modelled in the nephrocyte via the endocytosis assays. Proximal tubule cells are never mentioned in the paper and that needs addressed, the nephrocyte is not just a model of podocytes. It is unclear who these findings from the fly model relate to podocytes alone.

We added a paragraph in the discussion addressing this point.

14. There is persistent reference to 'cell injury' yet there is no insult to the cells that constitutes an injury as such; injury is due to gene over-expression that modulate components of the slit diaphragm. An injury model would align more with transient chemical or hydrostatic pressure provocations. Perhaps avoid using the term injury.

We agree that the term injury might be misleading and therefore changed it to 'diseased nephrocytes' throughout the manuscript. All changes are marked in red.

15. Mechanical force is mentioned in the discussion yet there is no methodological approach that addressed forces being exerted on nephrocytes nor the effect this had on nephrocyte function or phenotype. Without such a provocation, it is hard to state that the modifications to nephrocytes were linked to mechanotransduction.

As mentioned above, we performed additional experiments inducing hypo-osmotic stress and could show, that nephrocytes respond to these changes in their physical environment by swelling, but our approach did not reveal morphological changes. However, as outlined above, we used fly strains expressing the mechanosensor region domain of Cheerio, hence we kept the term mechanotransduction.

16. Figures 4 and 5 have no images of nephrocytes to support the quantified data. Can these be presented?

We added images for Figure 4 as Supplementary Figure 5. Images to support the quantification of the data shown in Figure 5 are now shown in Supplementary Figure 6.

June 17, 2022

Re: Life Science Alliance manuscript #LSA-2021-01281-TR

Dr. Sybille Koehler
Universität Hamburg
Martinistr. 52
Hamburg 20257
Germany

Dear Dr. Koehler,

Thank you for submitting your revised manuscript entitled "Drosophila Filamin exhibits a mechano-protective role in nephrocytes via Yorkie mediated hypertrophy" to Life Science Alliance. The manuscript has been seen by the original reviewers whose comments are appended below. While the reviewers continue to be overall positive about the work in terms of its suitability for Life Science Alliance, some important issues remain.

Please address the final Reviewer 1 minor points and the points raised by Reviewer 2 regarding the change in the language (and in the title as well) throughout the manuscript to better reflect the conclusions.

Our general policy is that papers are considered through only one revision cycle; however, given that the suggested changes are relatively minor, we are open to one additional short round of revision. Please note that I will expect to make a final decision without additional reviewer input upon resubmission.

Please submit the final revision within one month, along with a letter that includes a point by point response to the remaining reviewer comments.

To upload the revised version of your manuscript, please log in to your account: <https://lsa.msubmit.net/cgi-bin/main.plex>
You will be guided to complete the submission of your revised manuscript and to fill in all necessary information.

B. MANUSCRIPT ORGANIZATION AND FORMATTING:

Sincerely,

Reviewer #1 (Comments to the Authors (Required)):

The authors have provided significant additional data and analysis that greatly improve the insights from the manuscript. It is, in my assessment, ready for publication. I have just a few minor comments regarding the author's responses and revised manuscript.

a) It is commendable that the authors attempted the experiments examining the protective role of Cher. As the authors note, however, the severity of the morphological defects caused by depletion of Sns or Duf preclude their ability to detect an enhancement when Cher is also depleted. Thus, I agree that it is not particularly useful to include the data from those experiments, since their interpretation and insight are limited.

b) In response to my previous comment: "Figure 4 Can the data in panels F and G be merged together with D and E, respectively, to make comparisons between all genotypes more straightforward?"

The authors note that the protocols for these experiments differed in potentially meaningful ways, which makes direct comparisons difficult or inappropriate. I appreciate this and am fine with them leaving the figures as they originally had them.

c) The authors provide a nice set of additional experiments addressing the relationships between Cher, Hippo and TOR signaling. Based on these new data, they note in their response to reviewer comments: "Taken together, our data therefore shows, that TOR does not seem to be activated by Cheerio and does not seem to be a downstream target." While this is now reflected throughout the main text of the manuscript, some of the figure legends still seem to imply that TOR is involved in the Cher-mediated hypertrophy. For example, Figure 6 legend lists TOR as a mediator of Cher-activated hypertrophy. Similarly, the title of Figure 4 is: "TOR and Hippo signalling mediate the hypertrophy phenotype in nephrocytes". Since the previous figures and text describe the hypertrophy phenotype in the context of Cher activity, this seems to imply that TOR is mediating hypertrophy caused by Cher activity. Based on the new data, it seems the wording should be changed in these instances to focus on Yki/Hippo.

d) There are two incomplete sentences (lack subjects) in the new red text on pg.12:

"Implying no activation of TOR signalling upon expression of active Cheerio."

and

"Suggesting an inactivation of Hippo signalling, which is indicated by translocation of Yorkie into the nucleus, leading to the activation of downstream signalling pathways."

Referee Cross-Comments:

Having read the comments and responses for Reviewer#2, I agree with Reviewer#2 that framing this study as evidence of Cheerio's role in a mechano-protective response seems over-reaching. It is reasonable that the authors may present their thoughts in the Discussion as to how their data (e.g., the active vs inactive MSR variants) and previous studies fit into a model of Cheerio serving a mechano-protective role in nephrocytes, but the current set of experiments do not adequately support the more strongly worded conclusions in the paper (and title).

I also find the term "diseased nephrocytes" a bit confusing. Reviewer#2's suggestion of "abnormal" is good, or perhaps simply be explicit about it and refer to them as "nephrocytes depleted of Sns (or Duf)".

Reviewer #2 (Comments to the Authors (Required)):

The rebuttal is thorough (very thorough!) and addresses all the points I made. This is a great piece of work but it comes across as trying to be a pre-clinical model when it's really an excellent piece of basic science explaining a role for Filamin in insect nephrocytes.

A small change to the language would neither detract from the message or undermine the conclusions. Simply, the title and narrative needs to better reflect the evidence that Filamin maintains nephrocyte morphology via Yorkie - and that's OK to say in a title. The title at present overstates a near pre-clinical model of congenital nephrotic syndrome being corrected via a Filamin-dependent mechanotransduction signaling pathway - and that's simply not supported by the data.

I obviously still have reservations about the use of the term 'mechanotransduction' because there is no formal experiment demonstrating how a mechanical signal is transduced to elicit an outcome via the proposed mechanism, so the term 'mechanotransduction' is still being used inappropriately, in my opinion. The premise of this paper is nicely stated in the first line of the discussion - mechanical SHEAR forces causes podocyte injury - and yet the experiments with nephrocytes don't address that scenario.

To be honest, I'm not sure the manuscript would differ / suffer if the authors remove the term 'mechano-'. The term 'mechano-protective' can simply be 'protective'. There is nothing wrong with saying that Cheerio modulates TOR-dependent growth but it is misleading to say that this is due to a mechanosensation mechanism. The hypo-osmotic shock effect is interesting but doesn't really add anything in terms of demonstrating a mechanosensation phenotype - the cells just swell and overall morphology is

retained (I'm not entirely sure what else you were expecting). Swelling in response to hypo-osmotic solutions is not in itself an endogenously controlled cellular response mediated by filamin, its physics; you'd need to show some functional change, some signal linking the swelling to a filamin-dependent cellular response to the swelling or mechanical provocation.

The term 'injury' was accepted as potentially misleading, so it was changed to the term 'diseased' but the term 'injury' is still used in figure 6 AND in the title.

Also, the term 'diseased' is not better than injury because the cells are experimentally manipulated to be abnormal and then rescued from this state. 'Disease' or 'injury' would be accurate if an injury or disease were being modeled and that isn't the case - these are genetic manipulations (and not disease causing mutations) that are being conducted, its basic science and not a pre-clinical model. Again, its OK to not use the terms 'injury' or 'disease', just say 'abnormal', it doesn't detract from the paper's message. The hypothesis is that Cheerio may interact with slit diaphragm proteins to maintain nephrocyte biology, so knocking down or modulating potential interactors in that pathway is not causing injury or disease per se, you are simply testing that hypothesis. There is no need to use the terms 'injury' or 'disease'. By suggesting injury or disease it will be misconstrued that you think Filamin may be a clinical target for the treatment of congenital nephrotic syndromes.

Again, apologies if I've misunderstood anything. This is a really good paper and needs published - but the language, in my opinion, needs to be couched within a basic science narrative, rather than a pre-clinical narrative.

Reviewer #1 (Comments to the Authors (Required)):

The authors have provided significant additional data and analysis that greatly improve the insights from the manuscript. It is, in my assessment, ready for publication. I have just a few minor comments regarding the author's responses and revised manuscript.

a) It is commendable that the authors attempted the experiments examining the protective role of Cher. As the authors note, however, the severity of the morphological defects caused by depletion of Sns or Duf preclude their ability to detect an enhancement when Cher is also depleted. Thus, I agree that it is not particularly useful to include the data from those experiments, since their interpretation and insight are limited.

We thank the reviewer for this conclusion and will not add the rescue data to the final version of the manuscript.

b) In response to my previous comment: "Figure 4 Can the data in panels F and G be merged together with D and E, respectively, to make comparisons between all genotypes more straightforward?"

The authors note that the protocols for these experiments differed in potentially meaningful ways, which makes direct comparisons difficult or inappropriate. I appreciate this and am fine with them leaving the figures as they originally had them.

We thank the reviewer for this conclusion and will keep the data separate in the final version of the manuscript.

c) The authors provide a nice set of additional experiments addressing the relationships between Cher, Hippo and TOR signaling. Based on these new data, they note in their response to reviewer comments: "Taken together, our data therefore shows, that TOR does not seem to be activated by Cheerio and does not seem to be a downstream target." While this is now reflected throughout the main text of the manuscript, some of the figure legends still seem to imply that TOR is involved in the Cher-mediated hypertrophy. For example, Figure 6 legend lists TOR as a mediator of Cher-activated hypertrophy. Similarly, the title of Figure 4 is: "TOR and Hippo signalling mediate the hypertrophy phenotype in nephrocytes". Since the previous figures and text describe the hypertrophy phenotype in the context of Cher activity, this seems to imply that TOR is mediating hypertrophy caused by Cher activity. Based on the new data, it seems the wording should be changed in these instances to focus on Yki/Hippo.

We do apologize for this and changed the manuscript accordingly.

d) There are two incomplete sentences (lack subjects) in the new red text on pg.12: "Implying no activation of TOR signalling upon expression of active Cheerio."

and

"Suggesting an inactivation of Hippo signalling, which is indicated by translocation of Yorkie into the nucleus, leading to the activation of downstream signalling pathways."

We do apologize for this and changed the sentences accordingly.

Referee Cross-Comments:

Having read the comments and responses for Reviewer#2, I agree with Reviewer#2 that framing this study as evidence of Cheerio's role in a mechano-protective response seems over-reaching. It is reasonable that the authors may present their thoughts in the Discussion as to how their data (e.g., the active vs inactive MSR variants) and previous studies fit into a model of Cheerio serving a mechano-protective role in nephrocytes, but the current set of experiments do not adequately support the more strongly worded conclusions in the paper (and title).

We agree and changed our wording to 'protective' and kept our thoughts regarding the mechano-protective role of Cheerio in the discussion.

I also find the term "diseased nephrocytes" a bit confusing. Reviewer#2's suggestion of "abnormal" is good, or perhaps simply be explicit about it and refer to them as "nephrocytes depleted of Sns (or Duf)".

We agree and changed our wording throughout the manuscript.

Reviewer #2 (Comments to the Authors (Required)):

The rebuttal is thorough (very thorough!) and addresses all the points I made. This is a great piece of work but it comes across as trying to be a pre-clinical model when it's really an excellent piece of basic science explaining a role for Filamin in insect nephrocytes.

A small change to the language would neither detract from the message or undermine the conclusions. Simply, the title and narrative needs to better reflect the evidence that Filamin maintains nephrocyte morphology via Yorkie - and that's OK to say in a title. The title at present overstates a near pre-clinical model of congenital nephrotic syndrome being corrected via a Filamin-dependent mechanotransduction signaling pathway - and that's simply not supported by the data.

I obviously still have reservations about the use of the term 'mechanotransduction' because there is no formal experiment demonstrating how a mechanical signal is transduced to elicit an outcome via the proposed mechanism, so the term 'mechanotransduction' is still being used inappropriately, in my opinion. The premise of this paper is nicely stated in the first line of the discussion - mechanical SHEAR forces causes podocyte injury - and yet the experiments with nephrocytes don't address that scenario.

We agree and changed the wording to 'protective' throughout the manuscript and kept our thoughts about the mechano-protective role of Cheerio in the discussion.

To be honest, I'm not sure the manuscript would differ / suffer if the authors remove the term 'mechano-'. The term 'mechano-protective' can simply be 'protective'. There is nothing wrong with saying that Cheerio modulates TOR-dependent growth but it is misleading to say that this is due to a mechanosensation mechanism. The hypo-osmotic shock effect is interesting but doesn't really add anything in terms of

demonstrating a mechanosensation phenotype - the cells just swell and overall morphology is retained (I'm not entirely sure what else you were expecting). Swelling in response to hypo-osmotic solutions is not in itself an endogenously controlled cellular response mediated by filamin, its physics; you'd need to show some functional change, some signal linking the swelling to a filamin-dependent cellular response to the swelling or mechanical provocation.

We agree and changed the wording to 'protective' throughout the manuscript and kept our thoughts about the mechano-protective role of Cheerio in the discussion. We hoped to see a morphological phenotype after swelling of nephrocytes and would have aimed to show a rescue effect by simultaneous expression of active Cheerio, which would have suggested a potential mechano-protective role. As we did not observe a morphological phenotype within our experimental setting, we will now change the wording to only 'protective'.

The term 'injury' was accepted as potentially misleading, so it was changed to the term 'diseased' but the term 'injury' is still used in figure 6 AND in the title.

Also, the term 'diseased' is not better than injury because the cells are experimentally manipulated to be abnormal and then rescued from this state. 'Disease' or 'injury' would be accurate if an injury or disease were being modeled and that isn't the case - these are genetic manipulations (and not disease causing mutations) that are being conducted, its basic science and not a pre-clinical model. Again, its OK to not use the terms 'injury' or 'disease', just say 'abnormal', it doesn't detract from the paper's message. The hypothesis is that Cheerio may interact with slit diaphragm proteins to maintain nephrocyte biology, so knocking down or modulating potential interactors in that pathway is not causing injury or disease per se, you are simply testing that hypothesis. There is no need to use the terms 'injury' or 'disease'. By suggesting injury or disease it will be misconstrued that you think Filamin may be a clinical target for the treatment of congenital nephrotic syndromes.

We agree and changed our wording to either 'abnormal nephrocytes' or by just simply saying 'depletion of Duf and Sns', as suggested by Reviewer 1.

Again, apologies if I've misunderstood anything. This is a really good paper and needs published - but the language, in my opinion, needs to be couched within a basic science narrative, rather than a pre-clinical narrative.

July 14, 2022

RE: Life Science Alliance Manuscript #LSA-2021-01281-TRR

Dr. Sybille Koehler
Universität Hamburg
Martinistr. 52
Hamburg 20257
Germany

Dear Dr. Koehler,

Thank you for submitting your revised manuscript entitled "A protective role for Drosophila Filamin in nephrocytes via Yorkie mediated hypertrophy". We would be happy to publish your paper in Life Science Alliance pending final revisions necessary to meet our formatting guidelines.

- please add your figure legend section (including the supplementary figure legends) to the main manuscript text
- please use the [10 author names, et al.] format in your references (i.e. limit the author names to the first 10)
- please add a callout for Figure 5D, Figure S1D, Figure S2B-E, Figure S4C-D; please double-check your figure legend for Figure 3 and add panel F to the legend

Figure Check:

- Figure S7 legend is not complete. Please expand it.

A. FINAL FILES:

B. MANUSCRIPT ORGANIZATION AND FORMATTING:

Sincerely,

July 18, 2022

RE: Life Science Alliance Manuscript #LSA-2021-01281-TRRR

Dr. Sybille Koehler
Universität Hamburg
Martinistr. 52
Hamburg 20257
Germany

Dear Dr. Koehler,

Thank you for submitting your Research Article entitled "A protective role for Drosophila Filamin in nephrocytes via Yorkie mediated hypertrophy". It is a pleasure to let you know that your manuscript is now accepted for publication in Life Science Alliance. Congratulations on this interesting work.

DISTRIBUTION OF MATERIALS:

Again, congratulations on a very nice paper. I hope you found the review process to be constructive and are pleased with how the manuscript was handled editorially. We look forward to future exciting submissions from your lab.

Sincerely,
